# Pharmacologic interventions for postoperative nausea and vomiting after thyroidectomy: A systematic review and network meta-analysis

**Ye Jin Cho, Geun Joo Choi, Eun Jin Ahn, Hyun Kang**  *

Department of Anesthesiology and Pain Medicine, Chung-Ang University College of Medicine, Seoul, Republic of Korea

* roman00@naver.com

**Data Availability Statement:** All relevant data are within the paper and its Supporting Information files.

## Abstract

### Objective

To determine the effectiveness of pharmacologic interventions for preventing postoperative nausea and vomiting (PONV) in patients undergoing thyroidectomy.

### Design

Systematic review and network meta-analysis (NMA).

### Data sources

MEDLINE, EMBASE, Cochrane Central Register of Controlled Trials, and Google Scholar.

### Eligibility criteria, participants, and interventions

Randomized clinical trials that investigated the efficacy of pharmacologic interventions in preventing PONV in patients undergoing thyroidectomy were included. The primary endpoints were the incidences of postoperative nausea and vomiting (PONV), postoperative nausea (PON), postoperative vomiting (POV), use of rescue antiemetics, and incidence of complete response in the overall postoperative phases. The secondary endpoints were the same parameters assessed in the early, middle, and late postoperative phases. The surface under the cumulative ranking curve (SUCRA) values and rankograms were used to present the hierarchy of pharmacologic interventions.

### Results

Twenty-six studies (n = 3,467 patients) that investigated 17 different pharmacologic interventions were included. According to the SUCRA values, the incidence of PONV among the overall postoperative phases was lowest with propofol alone (16.1%), followed by palonosetron (27.5%), and with tropisetron (28.7%). The incidence of PON among the overall postoperative phases was lowest with propofol alone (11.8%), followed by tropisetron and propofol combination (14%), and ramosetron and dexamethasone combination (18.0%). The

**Funding:** This research was supported by the Basic Science Research Program through the National Research Foundation of Korea (NRF), which is funded by the Ministry of Education, Science, and Technology (2018R1A2A2A05021467). The funders had no role in study design, data collection and analysis, decision to publish, or preparation of the manuscript.

**Competing interests:** The authors have declared that no competing interests exist.

incidence of POV among the overall postoperative phases was lowest with tropisetron and propofol combination (2.2%), followed by ramosetron and dexamethasone combination (23.2%), and tropisetron alone (37.3%). The least usage of rescue antiemetics among the overall postoperative phases and the highest complete response was observed with tropisetron and propofol combination (3.9% and 96.6%, respectively).

## Conclusion

Propofol and tropisetron alone and in combination, and the ramosetron and dexamethasone combination effectively prevented PONV, PON, POV in patients undergoing thyroidectomy, with some heterogeneity observed in this NMA of full-text reports. Their use minimized the need for rescue antiemetics and enhanced the complete response.

## Trial registration number

CRD42018100002.

## Introduction

Postoperative nausea and vomiting (PONV) are the most common and unpleasant complications after anesthesia induction and surgery, and could result in aspiration pneumonia, fluid and electrolyte imbalances, and esophageal rupture [1–3]. Moreover, PONV prolongs the patients' length of hospital stay, increases healthcare costs, and decreases patient satisfaction [4–6]. In particular, vomiting after thyroidectomy may increase the incidence and severity of postsurgical complications, such as surgical wound dehiscence, postoperative hemorrhage, or neck hematoma, and in the worst case, airway obstruction might occur due to hematoma [7, 8].

The overall incidence of PONV has been reported to range from 22–52% after general anesthesia induction [9, 10]. However, the incidence of PONV after thyroidectomy increased to 60–84% when no prophylactic antiemetic is given [2, 11, 12], as surgical handling of neck during thyroidectomy induces intense vagal stimulation, and patients receiving thyroidectomy are mostly young or middle-aged women, in whom the risk of PONV is high [2].

Thus, numerous pharmacologic interventions, including antihistamines, anticholinergics, corticosteroids, and other multimodal approaches, have been studied for the prevention of PONV after thyroidectomy [8, 13–17]. However, the findings of these studies are conflicting and variable.

Although a few systematic reviews and meta-analyses have demonstrated the efficacy of dexamethasone to treat PONV after thyroidectomy [18–20], these studies focused only on the use of dexamethasone and compared only two groups. Thus, the relative efficacy of pharmacologic interventions remains unknown. Furthermore, these studies include those conducted before 2014. Recently, newer pharmacologic interventions and methodologies have been developed to prevent PONV after thyroidectomy, and large-scale high-impact studies have been published. Systematic reviews incorporating network meta-analyses (NMAs) can provide information on the hierarchy of competing interventions in terms of treatment rankings [21].

Therefore, we aimed to conduct a systematic review of randomized controlled trials (RCTs) and conduct an NMA to assess the efficacy of pharmacologic interventions used to prevent PONV in patients undergoing thyroidectomy. We believe that this study will provide insight into the treatment hierarchy of the different interventions.

## Materials and methods

### Protocol and registration

We developed the protocol for this systematic review and NMA according to the Preferred Reporting Items for Systematic Reviews and Meta-analyses (PRISMA) protocol statement [22] and registered it with the International Registration of Prospective Systematic Reviews (PROS-PERO network); registration number: CRD42018100002; accessible at (https://www.crd.york.ac.uk/PROSPERO/display_record.php?RecordID=100002), and published in a peer-reviewed journal [23].

This systematic review and NMA of pharmacologic interventions to prevent PONV after thyroidectomy was performed according to the protocol recommended by the Cochrane Collaboration [24] and reported according to the PRISMA extension for NMA guidelines [21].

### Inclusion criteria

We included only the RCTs that compared the efficacy of two or more pharmacologic interventions, or their combinations, to prevent PONV after thyroidectomy.

The PICO-SD information was as follows:

1. **Population (P):** (1) patients who underwent elective ambulatory thyroidectomy under general anesthesia; and (2) those who were given prophylactic medications for nausea and vomiting

2. **Intervention (I):** pharmacologic interventions to prevent PONV, including various 5-HT$_3$-receptor antagonists (ondansetron, ramosetron, palonosetron, granisetron, and dolasetron); corticosteroids (dexamethasone, etc.); lidocaine, midazolam, propofol, and other drugs alone or in combination with other pharmacologic agents, which is administered preoperative or intraoperative time period. If a drug was administered in different doses or different time of administration, it was regarded as same intervention.

3. **Comparison (C):** other pharmacologic interventions and/or their combination/s with other pharmacologic agents, placebo, or no treatment, which is administered preoperative or intraoperative time period. If a drug was administered in different doses or different time of administration, it was regarded as same intervention.

4. **Outcomes (O):** The primary endpoints were the incidences of postoperative nausea and vomiting (PONV), postoperative nausea (PON), postoperative vomiting (POV), use of rescue antiemetics, and the incidence of complete response (CR) in the overall postoperative phases. The secondary endpoints were PONV, PON, POV, use of rescue antiemetics, and the incidence of complete response in the early, middle, and late postoperative phases, and safety issues, including complications such as headache, dizziness, drowsiness, and constipation.

   The postoperative period was divided into the early, middle, late, and overall phases. The early phase was defined as 0–6 h postoperatively; middle phase, 6–24 h postoperatively; and late phase, more than 24 h postoperatively. If a study reported data at multiple time points within the same phase, data from the first time point were selected as the outcome of interest (e.g., if the study reported data at 0 h, 2 h, 4 h, and 6 h postoperatively, we only included the data at 0 h as the early phase). If the reported study data had overlapping time points between the phases, the data were classified into the phase containing a greater proportion of the overlapped range of time (e.g., if the study reported the data at 0–2 h and 2–24 h, we defined the data at 0–2 h as the early phase and that at 2–24 h as the middle phase). To ensure the inclusion of maximum number of studies, any PON, POV, and PONV data

from studies that do not mention a specific time point, as long as data were reported, were defined as the overall phase.

5. **Study design (SD):** peer-reviewed, randomized clinical studies.

### Exclusion criteria

1. Review articles, case reports, case series, letters to the editor, commentaries, proceedings, laboratory science studies, and other similar article types.

2. Studies that compared non-pharmacological interventions, such as the administration of oxygen, fluids, acupuncture, or regional blocks.

3. Studies that failed to report the outcomes of interest.
   No language or date restriction was applied.

### Information sources and search strategy

We searched MEDLINE, EMBASE, Cochrane Central Register of Controlled Trials (CENTRAL), and Google Scholar using the search terms related to pharmacologic interventions to prevent PONV after thyroidectomy from inception to Jun 15, 2020. Search terms used for MEDLINE and EMBASE are presented in the **S1 Search Term**. The references were imported to Endnote software 8.1 (Thompson Reuters, CA, USA) and duplicate articles were removed. Additional but relevant articles were identified by scanning the reference lists of articles obtained from the original search.

### Study selection

Two investigators (Choi GJ and Cho YJ) screened the titles and abstracts of the retrieved articles to identify RCTs meeting the abovementioned inclusion criteria. For the articles that were eligible based on their title or abstract, full paper was retrieved and evaluated. Potentially relevant studies chosen by at least one investigator were also retrieved and evaluated. To minimize data duplication due to multiple reporting, papers from the same author, organization, or country were compared. Articles meeting the inclusion criteria were assessed separately by two independent investigators, and any disagreements were resolved through mutual discussion. In cases where a consensus could not be reached, the dispute was resolved with the help of a third investigator (Kang H).

The degree of agreement between the two investigators (Choi GJ and Cho YJ) for study selection was computed using kappa statistics to measure the difference between the observed and expected agreements between them; i.e., whether they were selected at random or by chance only. Kappa values were interpreted as follows: (1) less than 0: less than chance agreement; (2) 0.01–0.20: slight agreement; (3) 0.21–0.40: fair agreement; (4) 0.41–0.60: moderate agreement; (5) 0.61–0.80: substantial agreement; and (6) 0.8–0.99: almost perfect agreement [25].

### Data extraction

Using a standardized extraction form, the following data were extracted independently by two investigators (Cho YJ and Ahn EJ): (1) title; (2) authors; (3) name of journal; (4) publication year; (5) study design; (6) competing interests; (7) country; (8) risk of bias; (9) number of patients in study; (10) types and doses of drugs compared; patients' (11) sex; (12) age; (13) weight; (14) height; (15) duration of anesthesia; (16) American Society of Anesthesiologists' physical status score; (17) inclusion criteria; (18) exclusion criteria; (19) type of surgery; (20)

type of anesthesia; (21) number of cases of PON, POV, and PONV overall and during the early, middle, and late postoperative phases; (22) the need for rescue antiemetics; and (23) number of cases of complete response.

If information was inadequate or missing, attempts were made to contact the study authors for additional information. If unsuccessful, efforts were made to obtain the missing information from the available data or was extracted from figures using the open source software, Plot Digitizer (version 2.6.8; http://plotdigitizer. sourceforge.net).

The reference lists were divided and distributed between two investigators for data extraction. The data extraction forms were created and cross-checked to verify the accuracy and consistency of the extracted data. Any disagreements were resolved through mutual discussion or with the help of a third investigator (Kang H).

## Study quality assessment

The quality of the studies was independently assessed by two study authors (Cho YJ and Ahn EJ), using version 2 of the Cochrane risk of bias tool for randomized trials (RoB 2) [4]. The risk of bias was evaluated by considering the following five potential sources of bias: (1) bias arising from the randomization process; (2) bias due to deviations from intended interventions; (3) bias due to missing outcome data; (4) bias in outcome measurements; and (5) bias in selection of the reported results. Thereafter, we evaluated an overall risk of bias judgment according to the domain-level judgments. The methodology for each domain was graded as "Low risk of bias," "Some concerns," and "High risk of bias," which reflected a low risk of bias, some concerns, and a high risk of bias, respectively [4].

## Statistical analysis

Ad-hoc tables were designed to summarize data from the included studies and show their key characteristics and any important question related to the aim of this review. If a trial result was reported with zero events in one group, then the event rate was artificially inflated by adding 0.5 to the events and total number of each group.

A multiple treatment comparison NMA is a meta-analysis generalization method that includes both direct and indirect RCT comparison of treatments. A random-effects NMA based on a frequentist framework was performed using STATA software (version 15; Stata-Corp LP, College Station, TX) based on *mvmeta* with NMA graphical tools developed by Chaimani and colleagues [26].

Before conducting the NMA, we determined whether a meta-analysis was possible. For this, we evaluated the transitivity assumptions. The transitivity assumption for whole network was assessed by visual comparing the distribution of potential effect modifier across comparisons such as patient eligibility criteria, demographics and types of pharmacologic interventions, study design, risk of bias (all risk versus removing "high risks of bias" for bias arising from the randomization process, and bias in measurement of the outcome) [27] (**S1 Table**).

A network plot linking all the included pharmacologic agents and their combinations with other pharmacologic agents was formed to indicate the types of pharmacologic agents, the number of patients who used them, and the level of pair-wise comparisons. In the network plot, nodes show the pharmacologic agents being compared and edges show the available direct comparisons between them. The nodes and edges were weighed on the basis of the number of patients and inverse values of standard errors of effect, respectively.

We evaluated the consistency assumption for the entire network using the design-by-treatment interaction model [28]. We also evaluated each closed loop in the network to evaluate local inconsistencies between the direct and indirect effect estimates for the same comparison.

For each loop, we estimated the inconsistency factor (IF) as the absolute difference between the direct and indirect estimates for each paired comparison in the loop [29].

Mean summary effects with confidence intervals (CIs) were presented together with their predictive intervals (PrIs) to facilitate interpretation of the results based on the magnitude of heterogeneity. PrIs is a kind of prediction interval. Prediction interval represents an estimate of an interval in which true effect size of future study will lie, with a certain probability, given what has already been observed, and account for heterogeneity. Prediction intervals are used in both frequentist statistics (predictive interval) and Bayesian statistics (credible interval) [30–32]. Thus, 95% PrIs represents an interval in which the future observation will fall with 95% certainty given observed sample from normal distribution.

Rankograms and cumulative ranking curves were generated for each pharmacologic agent. The rankogram plots are the probabilities for treatments to assume a possible rank. It is the probability that a given treatment ranks first, second, third, etc., among all the treatment agents evaluated in the NMA. We used the surface under the cumulative ranking curve (SUCRA) values to present the hierarchy of pharmacologic agents for the incidences of PON, POV, PONV, use of rescue antiemetics, and the incidence of complete response among the overall phases. SUCRA is a relative ranking measure that accounts for the uncertainty in the treatment order, i.e., it accounts for both the location and variance of all relative treatment effects [33]. A higher SUCRA value is regarded as a better result for an individual intervention. When ranking treatments, the closer the SUCRA value is to 100%, the higher is the treatment ranking relative to all the other treatments.

A comparison-adjusted funnel plot was generated to assess the presence of small-study effects [34].

## Results

### Study selection

From the search of MEDLINE, EMBASE, CENTRAL, and Google Scholar databases, 86 studies met the inclusion criteria and were included for further evaluation. A subsequent manual search retrieved 15 additional studies. Of these 101 articles, 7 studies were excluded because those were duplicated. Then, 45 were excluded after reviewing their titles and abstracts because they did not align with our objective. The full texts of the remaining 49 studies were reviewed in detail; 23 studies were excluded for the following reasons: study protocol [35], retrospective study design [36], study retraction [37, 38], non-reporting of the outcomes of interest [14, 39], non-reporting of comparison of interests [8, 40–51], and comparison with non-pharmacological interventions [52–55].

Thus, a total of 26 studies (a total of 3,467 patients) that included 17 different pharmacologic interventions were included in this NMA (**Fig 1**). The kappa value for the selected articles between the two reviewers was 0.844.

### Study characteristics

The characteristics of the 26 studies are summarized in **Table 1**. All the studies were performed in accordance with American Society of Anesthesiologists physical status classifications I, II, and III. These 26 studies were conducted in various countries, such as Greece [56, 57], China [17, 58], Belgium [12], Republic of Korea [13, 15, 16, 59–63], Norway [64], Portugal [65], Italy [66], Germany [67, 68], Turkey [8, 69], Japan [70, 71], Taiwan [72, 73], Switzerland [74], and Finland [75]. One study was published in Chinese, and the rest were published in English. Seventeen pharmacologic interventions, including ondansetron (Ond) [56, 59, 75], palonosetron (Pal) [59, 60], propofol (Pro) [12, 71], intralipid (Int) [12], granisetron (Gra) [56, 57, 62], tropisetron (Tro) [8, 17, 56, 57, 75], dexamethasone (Dex) [15, 17, 58, 61, 64–66, 68–70, 72–74],

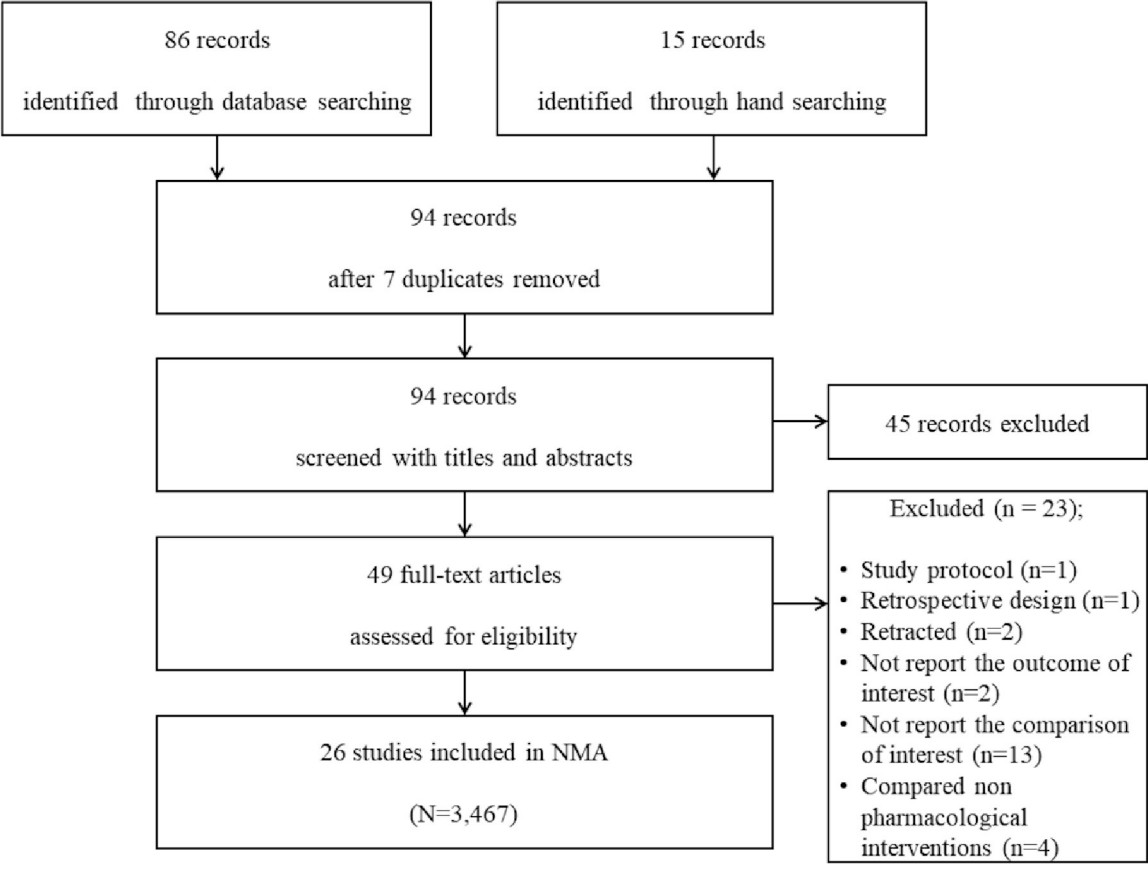

**Fig 1. PRISMA flowchart of included and excluded trials.**

tropisetron (Tro)+dexamethasone (Dex) [19], tropisetron (Tro)+propofol (Pro) [8], palonosetron (Pal)+dexamethasone (Dex) [60], ramosetron (Ram) [13, 15, 16, 61–63], ramosetron (Ram)+Dexamethasone (Dex) [15, 63], droperidol (Dro) [67, 71, 72], midazolam (Mid) [13, 67], dexamethasone (Dex)+Oral ginger (Gin) [69], ramosetron (Ram)+midazolam (Mid) [13] and metoclopramide (Met) [71, 75] were evaluated. Additional drugs used postoperatively were analgesics and antiemetics.

## Study quality assessment

Table 2 presents the risk of bias assessment for the included studies using the RoB2.

## Synthesis of results

For all outcomes of each datum, we presented the network plot (Fig 2), and expected mean ranking and pharmacologic agent SUCRA values for the outcomes (Fig 3). Inconsistency plot (S-Fig 4 in S1 File), CI and/or PrI plot compared with placebo (S-Fig 5 in S1 File), CI and/or PrI plot (S-Fig 6 in S1 File), rankogram (S-Fig 7 in S1 File), cumulative ranking curve (S-Fig 8 in S1 File), and comparison-adjusted funnel plot (S-Fig 9 in S1 File) are presented in the **S1 File**. Only the results for the primary end point, i.e., overall phase data are presented here; results for early, middle, and late phases are presented in the **S2 File**. The summary of the results is presented in S-Figs 2–9 in S1 File (Fig A, B, C, D and E correspond to PONV, PON, POV, use of rescue anti-emetics and complete response, respectively).

**Table 1. Characteristics of the trials included in the meta-analysis.**

| Study (1st author, year) | Country | Interventions | Sample size | Anesthetic technique | Additional drug administration (post-operative) | Outcome measurement for meta-analysis |
|---|---|---|---|---|---|---|
| Moon YE, 2012 | Republic of Korea | Ond 8mg bolus and 16mg in IV PCA | 50 | Pro 1.5–2.5mg/kg, fentanyl 1–2 µg/kg, rocuronium 0.8mg/kg IV maintained with sevoflurane in nitrous oxide/oxygen | Analgesia: meperidine 25mg IV | Incidence of PON, POV, PONV. Use of anti-emetics |
| | | | | | | Severity of nausea |
| | | Pal 0.075mg IV | 50 | | Antiemetics: Met 10mg IV | Incidence of side-effects |
| Ewalenk P, 1996 | Belgium | Pro 0.1mg/kg/hr IV | 32 | Fentanyl 2 µg/kg, thiopentone 3-5mg/kg, atracurium 0.4–0.5mg/kg IV maintained with isoflurane and nitrous oxide in oxygen | Analgesia: Piritamide 0.25mg/kg IM | Incidence of PON, POV, PONV. Use of anti-emetics |
| | | | | | | Severity of PONV |
| | | 10% Int 0.1mg/kg/hr IV | 32 | | Antiemetics: Met 10mg IV | Sedation score |
| Metaxari M, 2011 | Greece | Pla 5mg IV | 50 | Pro 2-3mg/kg, fentanyl 2 µg/kg, cisatracurium 0.15mg/kg IV maintained with sevoflurane in oxygen | Analgesia: paracetamol 1mg IV, pethidine 0.5-1mg/kg IM | Incidence of PON, POV |
| | | Gra 3mg IV | 50 | | | |
| | | Ond 4mg IV | 51 | | Antiemetics: Met 10mg IV | Severity of nausea |
| | | Tro 5mg IV | 52 | | | |
| Zhou H, 2012 | China | Dex 8mg IV | 50 | Pro 1.5–2.5mg/kg, Mid 0.1–0.2mg/kg, fentanyl 1.0–2.0 µg/kg, atracurium 0.3–0.6mg/kg IV maintained with sevoflurane in oxygen | Analgesia: pethidine 25mg IM | Incidence of PON, POV |
| | | | | | | Use of anti-emetics. |
| | | | | | | Complete response. |
| | | Tro 5mg IV | 50 | | Antiemetics: Met 10mg, Tro 5mg IV | Postoperative pain |
| | | | | | | Severity of PONV |
| | | | | | | Postoperative pain intensity |
| | | Dex 8mg + Tro 5mg IV | 50 | | | Adverse events, complications |
| Park JW, 2012 | Republic of Korea | Pal 0.075mg IV | 41 | Lidocaine 40mg, Pro 2mg/kg, rocuronium 0.6mg/kg IV maintained with sevoflurane in oxygen | Analgesia: ketolorac 30mg IV | Incidence of PON, POV, PONV |
| | | | | | | Severity of PONV |
| | | Pal 0.075mg + Dex 4mg IV | 43 | | Antiemetics: Ond | Complete response |
| Jeon Y, 2010 | Republic of Korea | Ram 0.3mg IV | 60 | Pro 2mg/kg, rocuronium 1mg/kg IV maintained with isoflurane and nitrous oxide in oxygen | Analgesia: ketolorac 30mg IV | Incidence of PON, POV |
| | | | | | | Severity of PONV |
| | | Dex 8mg IV | 60 | | Antiemetics: Met 10mg, IV | Use of rescue antiemetics |
| | | Ram 0.3mg + Dex 8mg IV | 60 | | | Occurrence of adverse events |
| Doksrod S, 2012 | Norway | Dex 0.3mg/kg IV | 40 | Fentanyl, Pro, vecuronium IV maintained with desflurane and nitrous oxide in oxygen | Analgesia: fentanyl 0.5 µg/kg IV, oxycodone 5mg orally | Incidence of PONV |
| | | | | | | Severity of PONV |
| | | | | | | Use of rescue antiemetics or analgesics |
| | | Dex 0.15mg/kg IV | 40 | | Antiemetics: Met 20mg, Ond 4 mg IV | Occurrence of side effects |
| | | Pla | 40 | | | |
| Barros A, 2013 | Portugal | Dex 4mg IV | 17 | Fentanyl 2 µg/kg, Pro, cisatracurium 0.15mg/kg IV maintained with sevoflurane | Analgesia: Ketorolac 30mg or parecoxib 40mg IV | Severity of PON, POV |
| | | | | | | Use of the PCA pump |
| | | | | | | Pain intensity |
| | | Pla | 17 | | Antiemetics: Ond 4mg or Pro 20mg IV | Sedation and shivering scores |
| | | | | | | Use of rescue antiemetics or analgesics |
| Schietroma M, 2013 | Italy | Dex 8mg IV | 163 | Sodium thiopental 5mg/kg, atracurium 0.5mg/kg IV maintained with remifentanil 0.25 µg/kg/min, sevoflurane in oxygen | Analgesia: Ketorolac 30mg IV | Incidence of recurrent laryngeal nerve palsy |
| | | Pla | 165 | | Antiemetics: Ond 4mg IV | Use of rescue antiemetics or analgesics |

*(Continued)*

**Table 1.** (*Continued*)

| Study (1st author, year) | Country | Interventions | Sample size | Anesthetic technique | Additional drug administration (post-operative) | Outcome measurement for meta-analysis |
|---|---|---|---|---|---|---|
| Eberhar LH, 1999 | Germany | Dro 5–7.5mg IV | 78 | Fentanyl 4 µg/kg, methohexitone 1–1.5mg/kg (ASA I-II) or etomidate 0.1–0.3mg/kg (ASA III-IV), atracurium 0.5mg/kg IV maintained with nitrous oxide in oxygen | Analgesia: piritramide IV | Post-operative mood and well-being |
| | | | | | | Incidence of PON, POV |
| | | | | | | Impact of PONV on post-operative mood and well-being |
| | | Mid 5–7.5mg IV *5mg: body weight<70kg, 7.5mg: body weight≥70kg | 72 | | Antiemetics: Met 10mg, dimenhydrinate 1mg/kg IV | Use of rescue antiemetics or analgesics |
| Song YK, 2013 | Republic of Korea | Pla | 41 | Remifentanil 1µg/kg, Pro 1-2mg/kg, rocuronium 0.9mg/kg IV maintained with desflurane in oxygen | Analgesia: ketorolac 30mg IV | Incidence of PON, POV and PONV |
| | | | | | | Severity of PONV |
| | | Dex 10mg IV | 41 | | Antiemetics: Met 10mg IV | Use of rescue antiemetics |
| | | | | | | Severity of PAS |
| | | Ram 0.3mg IV | 41 | | | Post-operative pain (VAS) |
| Akin A, 2006 | Turkey | Tro 5mg IV | 35 | Fentanyl 1µg/kg, thiopental 6-7mg/kg, vecuronium 0.1mg/kg IV maintained with desflurane in nitrous oxide and oxygen | Analgesia: diclofenac 75mg IV | Post-operative pain (VAS) |
| | | | | | | Incidence of PON, POV |
| | | Tro 5mg + Pro 0.5mg/kg IV | 35 | | Antiemetics: Met 10mg IV | Use of rescue antiemetics |
| | | Pla | 35 | | | Complete response |
| Tarantino I, 2015 | Germany | Dex 8mg IV | 76 | Pro, remifentanil, rocuronium IV with | Analgesia: paracetamol 1g oral, metamizol 1g oral, morphine 1mg IV | Incidence of PON, POV |
| | | | | | | Severity of PONV |
| | | | | | | Severity of pain, length of stay |
| | | Pla | 76 | | Antiemetics: Dro 0.5mg, Ond 4mg IV | Severity of adverse events |
| Fujii Y, 2007 | Japan | Pla | 25 | Pro 2mg/kg, fentanyl 2µg/kg, vecuronium 0.1mg/kg IV maintained with sevoflurane in nitrous oxide and oxygen | Analgesia: indomethacin 50mg | Incidence of PON, POV |
| | | | | | | Severity of nausea |
| | | Dex 4mg IV | 25 | | | Post-operative pain |
| | | Dex 8mg IV | 25 | | | |
| Papadima A, 2013 | Greece | Gra 3mg IV | 45 | Pro 2mg/kg, remifentanil 1µg/kg, cisatracurium 0.2mg/kg IV, meperidine 1mg/kg IM maintained with sevoflurane in oxygen | Analgesia: parecoxib 40mg IV, meperidine 50mg IM, | Post-operative pain (VAS) |
| | | | | | | Incidence of PON, POV |
| | | | | | | Severity of PON, POVV |
| | | Tro 5mg IV | 40 | | Antiemetics: Met 10mg IV | Use of rescue antiemetics |
| | | Pla | 42 | | | Side effects |
| Lee DC, 2011 | Republic of Korea | Pla | 65 | Pro (target effect-site concentration of 2.5–3.5µg/ml), remifentanil (target effect site concentration of 2.5–3.5ng/ml) continuous infusion, rocuronium 0.6mg/kg IV | Analgesia: ketorolac 30mg IV | Incidence of PON, POV |
| | | Ram 0.3mg IV | 65 | | Antiemetics: Met 10mg IV | Severity of PONV |
| | | | | | | Use of rescue anti-emetics and analgesics |
| | | | | | | Complete response |
| | | | | | | Pain score |
| | | | | | | Side effects of antiemetics |

(*Continued*)

**Table 1.** (Continued)

| Study (1st author, year) | Country | Interventions | Sample size | Anesthetic technique | Additional drug administration (post-operative) | Outcome measurement for meta-analysis |
|---|---|---|---|---|---|---|
| Tavlan A, 2006 | Turkey | Dex | 60 | Pro 2-3mg/kg, fentanyl 1.5μg/kg, atracurium basilate 0.5mg/kg IV maintained with isoflurane in nitrous oxide and oxygen | Analgesia: fentanyl 25–50μg, tenoxicam IV | Incidence of PON, POV |
|  |  |  |  |  |  | Severity of PON, POV |
|  |  | Dex + Gin 0.5g oral | 60 |  | Antiemetics: Met 10mg IV | Use of rescue analgesics, antiemetics |
| Lee SY, 2002 | Republic of Korea | Pla | 41 | Thiopentone 5mg/kg, vecuronium 0.1mg/kg or succinylcholine 1–1.5mg/kg IV maintained with enflurane in nitrous oxide and oxygen | Antiemetics: Met 10mg IV or IM | Incidence of PON, POV, PONV |
|  |  |  |  |  |  | Severity of PONV |
|  |  | Gra 20μg/kg IV | 36 |  |  | Adverse events |
|  |  | Ram 4μg/kg IV | 36 |  |  | Use of rescue antiemetics |
| Wang JJ, 1999 | Taiwan | Dex 10mg IV | 38 | Pro 2–2.5mg/kg, fentanyl 2μg/kg, glycopyrrolate 0.2mg, vecuronium 0.15mg/kg IV maintained with isoflurane in oxygen | Analgesia: diclofenac 75mg IV | Incidence of PON, PONV |
|  |  | Dro 1.25mg IV | 40 |  | Antiemetics: Ond 4mg IV | Severity of PON |
|  |  | Pla | 38 |  |  | Post-operative pain (VAS) |
|  |  |  |  |  |  | Occurrence of sore throat, restlessness |
| Zhang HW, 2016 | China | Dex 0.1mg/kg IV | 103 | Pro 2mg/kg, fentanyl 4μg/kg, rocuronium bromide 0.6mg/kg IV, μg/kg | Analgesia: diclofenac 50mg IV | Incidence of PON, POV |
|  |  |  |  |  |  | Use of rescue anti-emetics |
|  |  |  |  |  |  | Post-operative pain (VAS) |
|  |  | Pla | 130 |  |  | Blood glucose level |
| Kim WJ, 2013 | Republic of Korea | Ram 0.3mg IV | 30 | Fentanyl 2μg/kg, thiopental 5mg/kg, rocuronium bromide 0.8mg/kg IV maintained with sevoflurane in nitrous oxide in oxygen | Analgesia: ketorolac 30mg IV | Incidence of POV |
|  |  |  |  |  |  | Severity of PON |
|  |  |  |  |  |  | Post-operative pain (VAS) |
|  |  | Mid 75μg/kg IV | 32 |  | Antiemetics: Met 10mg, Dex 5mg IV | Use of rescue anti-emetics |
|  |  | Ram 0.3mg + Mid 75μg/kg IV | 32 |  |  |  |
| Worni M, 2008 | Switzerland | Pla | 35 | Pro/thiopental, atracurium, isoflurane or sevoflurane and fentanyl 5–10μg/kg IV | Analgesia: metamizole or morphine 1g IV or SC | Incidence of PON, POV and PONV |
|  |  |  |  |  |  | Severity of PON |
|  |  | Dex 8mg IV | 37 |  | Antiemetics: Ond 4mg, Dro 0.625mg IV | Post-operative pain (VAS) |
|  |  |  |  |  |  | Voice function |
|  |  |  |  |  |  | Severity of use of rescue anti-emetics, analgesics |
| Wang JJ, 2000 | Taiwan | Dex10mg | 44 | Pro 2.5mg/kg, glycopyrrolate 0.2mg, fentanyl 2μg/kg, vecuronium 0.15mg/kg IV maintained with isoflurane in oxygen | Analgesia: diclofenac 75mg IM | Incidence of PON, POV |
|  |  |  |  |  |  | Severity of PON, POV |
|  |  |  |  |  |  | Use of rescue antiemetics, analgesics |
|  |  | Dex 5mg | 43 |  |  | Complete response |
|  |  |  |  |  |  | Post-operative pain (VAS) |
|  |  | Dex 2.5mg | 43 |  | Antiemetics: Ond 4mg IV | Side effects |
|  |  | Dex 1.25mg | 44 |  |  |  |
|  |  | Pla | 43 |  |  |  |

(*Continued*)

**Table 1.** (Continued)

| Study (1st author, year) | Country | Interventions | Sample size | Anesthetic technique | Additional drug administration (post-operative) | Outcome measurement for meta-analysis |
|---|---|---|---|---|---|---|
| Fujii Y, 2001 | Japan | Pro 0.5mg.kg IV | 30 | Thiopentone 5mg/kg, fentanyl 2µg/kg, vecuronium 0.2mg/kg IV maintained with sevoflurane in nitrous oxide and oxygen | Analgesia: indomethacin 50mg rectally, | Incidence of PON, POV, PONV |
| | | | | | | Severity of PON |
| | | | | | | Sedation score |
| | | Dro 20µg/kg IV | 30 | | Antiemetics: perphenazine IV | Use of rescue antiemetics |
| | | Met 0.2mg/kg IV | 30 | | | |
| Jokela R, 2002 | Finland | Ond 16mg IV | 60 | Glycopyrrolate 0.2mg, fentanyl 2–3µg/kg, Pro 2-3mg/kg, rocuronium 0.5mg/kg IV maintained with sevoflurane in oxygen | Analgesia: oxycodone 0.05mg/kg IV or 0.1mg/kg IM, paracetamol 1g | Incidence of PON, PONV |
| | | | | | | Severity of PONV |
| | | | | | | Use of rescue antiemetics, analgesics |
| | | Tro 5mg IV | 60 | | Antiemetics: Dro 0.75mg IV | Post-operative pain (VAS) |
| | | Met 10mg IV | 59 | | | Incidence of adverse events |
| Lee MJ, 2015 | Republic of Korea | Pla | 36 | Pro 1-2mg/kg, remifentanil 1µg/kg IV maintained with desflurane in oxygen | Analgesia: ketorolac 30mg IV | Incidence of PON, POV |
| | | | | | | Severity of PON, POV |
| | | Ram 0.3mg | 36 | | Antiemetics: Met 10mg IV | Post-operative pain (VAS) |
| | | Ram 0.3mg + Dex 5mg | 36 | | | Incidence of adverse events |
| | | | | | | Use of rescue antiemetics, analgesics |

PONV: post-operative nausea and vomiting; IV: intravenous; Ond: ondansetron; Pal: palonosetron; PCA: patient-controlled analgesia; IM: intramuscular; Pla: placebo; Int: intralipid; Gra: granisetron; Tro: tropisetron; Dex: dexamethasone; Pro: proprofol; Dia: diazepam; Ram: ramosetron; Dro: droperidol; Mid: midazolam; VAS: visual analogue pain score; TCI: target-controlled infusion; PAS: postanesthetic shivering; TCI: target-controlled infusion; SC: subcutaneous; Met: metoclopramide; Gin: oral ginger

**Postoperative nausea and vomiting.** **Fig 2A** shows the network plot of the pharmacologic interventions comparing PONV in the overall phase. Ten pharmacologic interventions (Pro, Pal, Tro, Gra, Ond, Ram, Dro, Int, Dex, and Met) were compared in eight studies (857 patients) [12, 59, 61, 62, 71, 72, 74, 75].

The evaluation of network inconsistency using the design-by-treatment interaction model suggested a significant network inconsistency [$F_{(3,5)}$ = 3.87; P = 0.0897]. There were five closed loops in the network generated from the comparisons of PONV, but two loops (Ond-Tro-Met [75] and Pro-Dro-Met [71]) consisted of only multi-arm trials. Of the three closed loops, an inconsistency was observed in the 1-6-9 (Pla-Gra-Ram) loop [62] (S-Fig 4A in S1 File).

Treatment with Pro and Ram had lower incidences of PONV than Pla in the overall phase in terms of 95% CIs (S-Fig 5A, S-Fig 6A in S1 File and **Table 3**).

The rankogram and cumulative ranking plot showed that Pro had the lowest incidence of PONV in the overall phase (S-Fig 7A and S-Fig 8A in S1 File).

The SUCRA plot revealed that the incidence of PONV in the overall phase was lowest with Pro (16.1%), followed by Pal (27.5%), and with Tro (28.7%) (**Fig 3A**).

**Table 2. Risk of bias assessment.**

| Study (1st author, year) | Bias arising from the randomization process | Bias due to deviations from intended interventions | Bias due to missing outcome data | Bias in measurement of the outcome. | Bias in selection of the reported result | Overall risk of bias judgement |
|---|---|---|---|---|---|---|
| Moon YE, 2012 | Low risk | Low risk | Low risk | Low risk | Low risk | Low risk |
| Ewalenk P, 1996 | Some concerns | Low risk | Low risk | Low risk | Low risk | Some concerns |
| Metaxari M, 2011 | Some concerns | Low risk | Low risk | Low risk | Low risk | Some concerns |
| Zhou H, 2012 | Some concerns | Low risk | Low risk | Low risk | Low risk | Some concerns |
| Park JW, 2012 | Some concerns | Low risk | Low risk | Some concerns | Low risk | High risk |
| Jeon Y, 2010 | Low risk | Low risk | Low risk | Low risk | Low risk | Low risk |
| Doksrod S, 2012 | Low risk | Low risk | Low risk | Some concerns | Low risk | Some concerns |
| Barros A, 2013 | Some concerns | Low risk | Low risk | Low risk | Low risk | Some concerns |
| Schietrom M, 2013 | Low risk | Low risk | Low risk | Low risk | Low risk | Low risk |
| Eberhar LH, 1999 | Some concerns | Low risk | Low risk | Low risk | Low risk | Some concerns |
| Song YK, 2013 | Low risk | Low risk | Low risk | Some concerns | Low risk | Some concerns |
| Akin A, 2006 | Some concerns | Low risk | Low risk | Some concerns | Low risk | High risk |
| Tarantino I, 2015 | Low risk | Low risk | Low risk | Low risk | Low risk | Low risk |
| Fujii Y, 2007 | Some concerns | Low risk | Low risk | Low risk | Low risk | Some concerns |
| Papadima A, 2013 | Some concerns | Low risk | Low risk | Low risk | Low risk | Some concerns |
| Lee DC, 2011 | Some concerns | Low risk | Low risk | Low risk | Low risk | Some concerns |
| Tavlan A, 2006 | Low risk | Low risk | Low risk | Low risk | Low risk | Low risk |
| Lee SY, 2002 | Some concerns | Low risk | Low risk | Low risk | Low risk | Some concerns |
| Wang JJ, 1999 | Some concerns | Low risk | Low risk | Low risk | Low risk | Some concerns |
| Zhang HW, 2016 | Some concerns | Low risk | Low risk | Low risk | Low risk | Some concerns |
| Kim WJ, 2013 | Low risk | Low risk | Low risk | Low risk | Low risk | Low risk |
| Worni M, 2008 | Some concerns | Low risk | Low risk | Low risk | Low risk | Some concerns |
| Wang JJ, 2000 | Some concerns | Low risk | Low risk | Low risk | Low risk | Some concerns |
| Fujii Y, 2001 | Some concerns | Low risk | Low risk | Low risk | Low risk | Some concerns |
| Jokela R, 2002 | Low risk | Low risk | Low risk | Some concerns | Low risk | Some concerns |
| Lee MJ, 2015 | Low risk | Low risk | Low risk | Low risk | Low risk | Low risk |

Publication bias was less likely in the comparison-adjusted funnel plot (S-Fig 9A in S1 File).

**Postoperative nausea.** Thirteen pharmacologic interventions (Pro, Tro+Pro, Ram+Dex, Pal, Met, Ram, Gra, Dex+Gin, Tro, Ond, Dro, Dex, and Mid) were compared in 13 studies, including 1,676 patients (**Fig 2B**) [8, 15, 59, 61–63, 67, 69–73, 75].

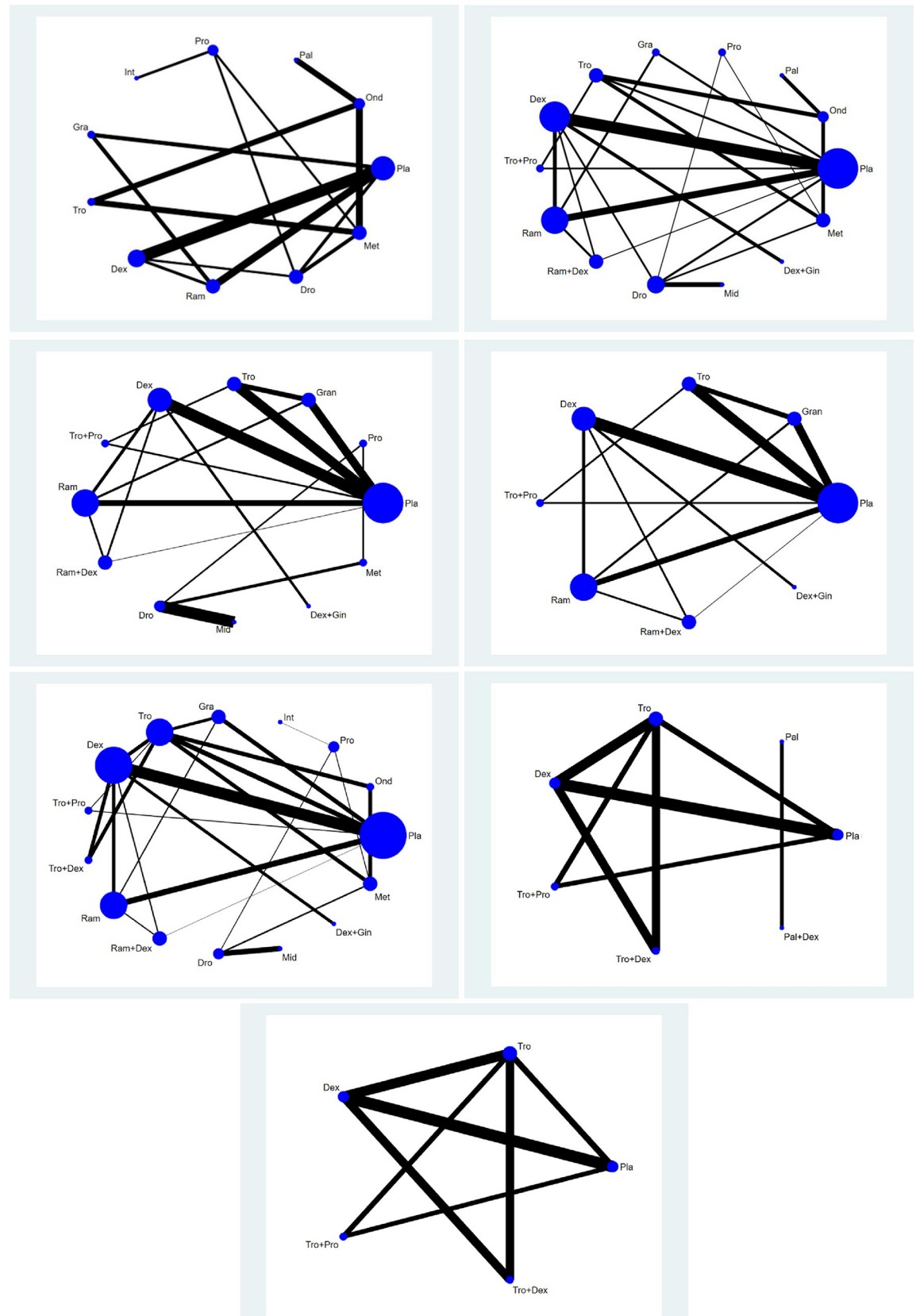

**Fig 2. Network plot of included studies comparing different pharmacological strategies. A: PONV, B:.PON, C1: POV, C2: POV excluding separate loops, D: use of rescue anti-emetics. E1: complete response, E2: complete response excluding separate loops.** The nodes show a comparison of pharmacological regimens to prevent PONV and the edges show the available direct comparisons among the pharmacological regimens. The nodes and edges are weighed on the basis of the number of included patients and inverse of standard error of effect.

The evaluation of the network inconsistency using the design-by-treatment interaction model suggested there was no evidences of statistically significant consistency [$F(7,9) = 2.90$ $P = 0.0698$].

There were 10 closed loops in the networks generated from the comparisons of postoperative nausea, but 3 loops (Pla(Placebo)-Tro-Tro+Pro [8], Ond-Tro-Met [75], Pro-Dro-Met [71]) consisted of only multi-arm trials. Although most loops showed no relevance in the local inconsistency between the direct and indirect point estimates, inconsistency was observed between the direct and indirect point estimates in the 1-5-9 loop (which included Pla-Gra-Ram) (S-Fig 4B in S1 File).

In terms of Cis, Pro, Tro+Pro, Ram+Dex, Met, Ram, and Dex showed lower incidences of mild PON than Pla among the overall phase (S-Fig 5B in S1 File).

Pro showed a lower incidence of PON than Dro and Mid; and Dex and Mid showed a higher incidence of PON in the overall phase than Tro+Pro and Ram+Dex (S-Fig 6B in S1 File).

The rankogram and cumulative ranking plot showed Pro to be the most effective pharmacologic intervention for reducing mild PON in the overall phase (S-Fig 7B, S-Fig 8B in S1 File and **Table 4**).

The SUCRA plots showed that the incidence of mild PON in the overall phase was lowest in Pro (11.8%), followed by Tro+Pro (14%), and Ram+Dex (18%) (**Fig 3B**).

The comparison-adjusted funnel plots suggested a less likely publication bias (S-Fig 9B in S1 File).

**Postoperative vomiting.**   Eleven studies (1,367 patients) measured the frequencies of postoperative vomiting. **Fig 2C1** shows the network graph of the 11 pharmacologic interventions (Tro+Pro, Ram+Dex, Tro, Ram, Gra, Dex+Gin, Dex, Pro, Dro, Mid and Met) that were compared in terms of POV in the overall phase after thyroidectomy [8, 15, 57, 61–63, 67, 69–71, 73].

As two studies (Dro vs. Mid [67] and Pro vs. Dro vs. Met [71]) were separated from the loops, the NMA was performed without them. Thus, a total of nine studies with a total of 1,127 patients were analyzed. **Fig 2C2** shows the network graph of the seven pharmacologic interventions (Tro+Pro, Ram+Dex, Tro, Ram, Gra, Dex+Gin, and Dex) that were compared in terms of POV in the overall phase after thyroidectomy [8, 15, 57, 61–63, 69, 70, 73].

The evaluation of the network inconsistency using the design-by-treatment interaction model suggested no significant inconsistency [$F(7,7) = 1.58$; $P = 0.2813$]. There were seven closed loops in the network generated from the comparisons of POV, which showed there was no evidence of significance in the local inconsistency between the direct and indirect point estimates (S-Fig 4C in S1 File).

Tro+Pro, Ram+Dex, Tro, Ram, and Gra showed a lower incidence of POV than Pla in the overall phase, which were significant only in terms of their 95% CIs, but not their 95% PrIs (S-Fig 5C in S1 File).

Non-significant data in terms of the 95% PrIs suggest that any future RCT could change the significance of the efficacy of these comparisons. Tro+Pro showed a lower incidence of POV in the overall phase than Gra, Tro, Dex, Ram, and Dex+Gin only in terms of their 95% CIs (S-Fig 6C in S1 File).

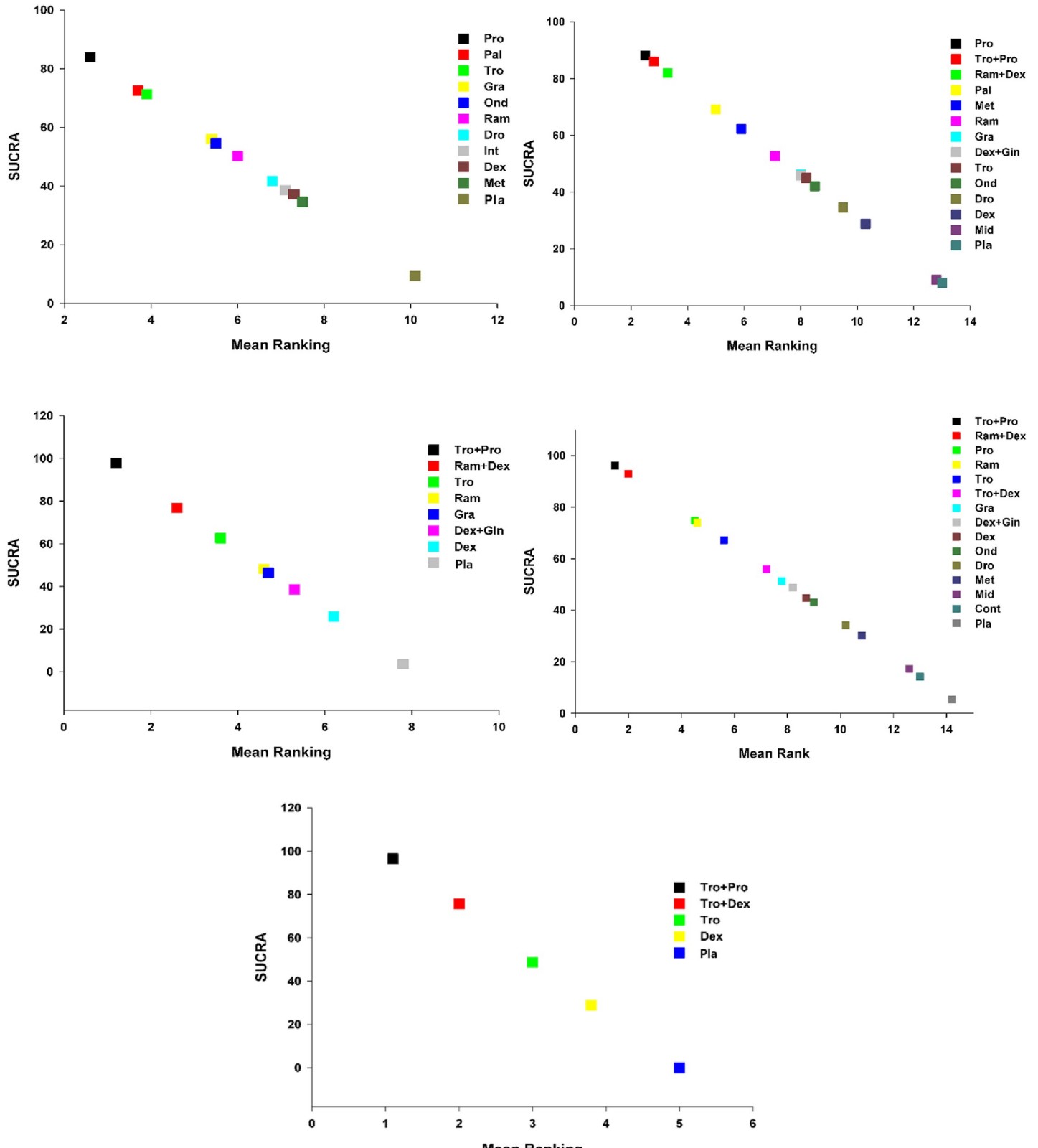

**Fig 3. Expected mean ranking and SUCRA values for PONV.** A. X-axis corresponds to expected mean ranking based on SUCRA (surface of under cumulative ranking curve) value, and Y-axis corresponds to SUCRA value. Fig 3B. Expected mean ranking and SUCRA values for PON. X-axis corresponds to expected mean ranking based

on SUCRA (surface of under cumulative ranking curve) value, and Y-axis corresponds to SUCRA value. Fig 3C. Expected mean ranking and SUCRA values for POV. X-axis corresponds to expected mean ranking based on SUCRA (surface of under cumulative ranking curve) value, and Y-axis corresponds to SUCRA value. Fig 3D. Expected mean ranking and SUCRA values for use of rescue anti-emetics. X-axis corresponds to expected mean ranking based on SUCRA (surface of under cumulative ranking curve) value, and Y-axis corresponds to SUCRA value. Fig 3E. Expected mean ranking and SUCRA values for complete response. X-axis corresponds to expected mean ranking based on SUCRA (surface of under cumulative ranking curve) value, and Y-axis corresponds to SUCRA value.

The rankogram showed that Tro+Pro had the lowest incidence of POV in the overall phase (S-Fig 7C in S1 File).

The cumulative ranking plot was drawn and the SUCRA probabilities of the different pharmacologic interventions for reducing POV in the overall phase were calculated (S-Fig 8C in S1 File, **Table 5**).

The expected mean rankings and the SUCRA values of each pharmacologic intervention are presented in **Fig 3C**.

According to the SUCRA values, the incidence of POV was lowest with Tro+Pro (2.2%), followed by Ram+Dex (23.2%), and with Tro (37.3%).

The comparison-adjusted funnel plots show that the funnel plots were symmetrical around the zero line, which suggested a less likely publication bias (S-Fig 9C in S1 File).

**Use of rescue antiemetics.** Fourteen pharmacologic interventions (Tro+Pro, Ram+Dex, Pro, Ram, Tro, Tro+Dex, Gra, Dex+Gin, Dex, Ond, Dro, Met, Mid, and Int) were compared in 17 studies (2,392 patients) [8, 12, 15, 17, 36, 57, 58, 61–63, 65–67, 69, 71, 73, 75] (**Fig 2D**).

**Table 3. League table for PONV.**

| Pro | 0.77 (0.02,23.96) | 0.62 (0.03,10.94) | 0.29 (0.01,8.74) | 0.34 (0.02,5.96) | 0.23 (0.01,5.36) | 0.18 (0.02,1.51) | 0.14 (0.02,1.18) | 0.14 (0.01,2.54) | 0.15 (0.02,1.32) | 0.05 (0.00,0.81) |
|---|---|---|---|---|---|---|---|---|---|---|
| 1.30 (0.04,40.68) | Pal | 0.80 (0.05,12.02) | 0.38 (0.01,27.04) | 0.44 (0.07,3.00) | 0.30 (0.01,17.54) | 0.23 (0.01,6.55) | 0.19 (0.00,10.57) | 0.18 (0.00,8.81) | 0.20 (0.01,2.94) | 0.06 (0.00,2.84) |
| 1.62 (0.09,28.85) | 1.25 (0.08,18.67) | Tro | 0.48 (0.01,21.68) | 0.55 (0.08,3.76) | 0.38 (0.01,13.71) | 0.29 (0.02,4.57) | 0.23 (0.01,8.24) | 0.23 (0.01,6.72) | 0.25 (0.04,1.69) | 0.08 (0.00,2.16) |
| 3.40 (0.11,100.83) | 2.61 (0.04,183.83) | 2.09 (0.05,94.85) | Gra | 1.16 (0.03,51.86) | 0.79 (0.12,5.08) | 0.60 (0.04,8.20) | 0.49 (0.01,26.40) | 0.48 (0.06,3.88) | 0.52 (0.02,14.19) | 0.16 (0.02,1.01) |
| 2.94 (0.17,51.35) | 2.25 (0.33,15.23) | 1.81 (0.27,12.30) | 0.86 (0.02,38.73) | Ond | 0.69 (0.02,24.48) | 0.52 (0.03,8.12) | 0.42 (0.01,14.71) | 0.41 (0.01,11.99) | 0.45 (0.07,2.98) | 0.14 (0.00,3.85) |
| 4.28 (0.19,98.07) | 3.28 (0.06,188.99) | 2.63 (0.07,95.12) | 1.26 (0.20,8.06) | 1.46 (0.04,51.97) | Ram | 0.75 (0.08,7.32) | 0.62 (0.01,26.75) | 0.60 (0.13,2.92) | 0.66 (0.03,13.70) | 0.20 (0.05,0.79) |
| 5.69 (0.66,48.94) | 4.36 (0.15,124.89) | 3.50 (0.22,56.01) | 1.67 (0.12,22.99) | 1.94 (0.12,30.49) | 1.33 (0.14,12.94) | Dro | 0.82 (0.04,16.57) | 0.80 (0.12,5.53) | 0.87 (0.12,6.52) | 0.26 (0.04,1.75) |
| 6.94 (0.85,56.68) | 5.33 (0.09,300.16) | 4.28 (0.12,150.64) | 2.04 (0.04,110.27) | 2.37 (0.07,82.29) | 1.62 (0.04,70.48) | 1.22 (0.06,24.69) | Int | 0.98 (0.03,34.92) | 1.07 (0.05,21.59) | 0.32 (0.01,11.23) |
| 7.08 (0.39,127.31) | 5.43 (0.11,260.03) | 4.36 (0.15,127.60) | 2.08 (0.26,16.84) | 2.41 (0.08,69.66) | 1.65 (0.34,7.99) | 1.24 (0.18,8.56) | 1.02 (0.03,36.28) | Dex | 1.09 (0.07,17.63) | 0.33 (0.11,1.02) |
| 6.50 (0.76,55.91) | 4.99 (0.34,73.15) | 4.00 (0.59,27.03) | 1.91 (0.07,51.96) | 2.21 (0.34,14.61) | 1.52 (0.07,31.63) | 1.14 (0.15,8.52) | 0.94 (0.05,18.93) | 0.92 (0.06,14.88) | Met | 0.30 (0.02,4.76) |
| 21.55 (1.23,378.18) | 16.54 (0.35,777.40) | 13.27 (0.46,380.43) | 6.35 (0.99,40.47) | 7.34 (0.26,207.66) | 5.04 (1.27,19.97) | 3.79 (0.57,25.10) | 3.10 (0.09,108.28) | 3.05 (0.98,9.43) | 3.32 (0.21,52.33) | Pla |

Dark gray: Comparison, Light gray: Column compared with row, White: Row compared with column. Data are RRs (95% CI) in the column-defining treatment compared with the row-defining treatment or row-defining treatment compared with the column-defining treatment. For column compared with row, RRs higher than 1 favour the column-defining treatment. For row compared to column, RRs lower than 1 favour the row-defining treatment. RR = risk ratio. CI = confidence interval. Pro: proprofol; Pal: palonosetron; Tro: tropisetron; Gra: granisetron; Ond: ondansetron; Ram: ramosetron; Dro: droperidol; Int: intralipid; Dex: dexamethasone; Met: metoclopramide; Pla: placebo

**Table 4. League table for PON.**

| | Pro | Tro+Pro | Ram+Dex | Pal | Met | Ram | Gra | Dex+Gin | Tro | Ond | Dro | Dex | Mid | Pla |
|---|---|---|---|---|---|---|---|---|---|---|---|---|---|---|
| **Pro** | **Pro** | 0.81 (0.07,9.37) | 0.65 (0.06,7.32) | 0.40 (0.03,5.53) | 0.29 (0.04,1.98) | 0.23 (0.02,2.17) | 0.20 (0.02,2.41) | 0.19 (0.01,2.43) | 0.19 (0.02,1.59) | 0.18 (0.02,1.69) | 0.14 (0.02,0.92) | 0.12 (0.01,1.07) | 0.05 (0.00,0.48) | 0.06 (0.01,0.50) |
| **Tro+Pro** | 1.24 (0.11,14.37) | **Tro+Pro** | 0.80 (0.11,5.63) | 0.50 (0.05,5.08) | 0.36 (0.06,2.13) | 0.29 (0.05,1.57) | 0.24 (0.03,1.86) | 0.24 (0.03,1.97) | 0.23 (0.05,1.04) | 0.22 (0.03,1.47) | 0.17 (0.03,1.03) | 0.15 (0.03,0.79) | 0.06 (0.01,0.54) | 0.07 (0.02,0.34) |
| **Ram+Dex** | 1.55 (0.14,17.54) | 1.25 (0.18,8.79) | **Ram+Dex** | 0.62 (0.05,7.17) | 0.45 (0.07,2.88) | 0.36 (0.11,1.19) | 0.30 (0.05,1.72) | 0.29 (0.05,1.81) | 0.29 (0.05,1.67) | 0.27 (0.04,2.12) | 0.21 (0.04,1.12) | 0.19 (0.06,0.64) | 0.07 (0.01,0.61) | 0.09 (0.03,0.32) |
| **Pal** | 2.48 (0.18,33.96) | 2.00 (0.20,20.36) | 1.60 (0.14,18.37) | **Pal** | 0.72 (0.11,4.59) | 0.57 (0.06,5.44) | 0.48 (0.04,6.00) | 0.47 (0.04,6.16) | 0.47 (0.07,2.93) | 0.44 (0.12,1.66) | 0.34 (0.04,2.92) | 0.30 (0.03,2.72) | 0.12 (0.01,1.45) | 0.15 (0.02,1.25) |
| **Met** | 3.46 (0.51,23.65) | 2.79 (0.47,16.61) | 2.23 (0.35,14.37) | 1.39 (0.22,8.93) | **Met** | 0.80 (0.16,3.99) | 0.68 (0.09,4.82) | 0.66 (0.09,4.97) | 0.65 (0.20,2.13) | 0.61 (0.17,2.23) | 0.48 (0.13,1.77) | 0.42 (0.09,1.93) | 0.17 (0.03,1.04) | 0.21 (0.05,0.87) |
| **Ram** | 4.33 (0.46,40.68) | 3.49 (0.64,19.23) | 2.80 (0.84,9.34) | 1.75 (0.18,16.58) | 1.25 (0.25,6.25) | **Ram** | 0.85 (0.22,3.28) | 0.82 (0.17,4.01) | 0.82 (0.19,3.55) | 0.77 (0.12,4.71) | 0.60 (0.15,2.36) | 0.53 (0.23,1.23) | 0.21 (0.03,1.37) | 0.26 (0.12,0.56) |
| **Gra** | 5.12 (0.42,63.07) | 4.13 (0.54,31.77) | 3.31 (0.58,18.75) | 2.06 (0.17,25.58) | 1.48 (0.21,10.55) | 1.18 (0.30,4.59) | **Gra** | 0.97 (0.14,7.02) | 0.97 (0.15,6.13) | 0.91 (0.11,7.67) | 0.70 (0.12,4.20) | 0.63 (0.15,2.67) | 0.25 (0.03,2.22) | 0.31 (0.08,1.19) |
| **Dex+Gin** | 5.26 (0.41,67.19) | 4.24 (0.51,35.48) | 3.39 (0.55,20.81) | 2.12 (0.16,27.70) | 1.52 (0.20,11.49) | 1.21 (0.25,5.91) | 1.03 (0.14,7.40) | **Dex+Gin** | 0.99 (0.14,6.83) | 0.93 (0.10,8.38) | 0.72 (0.12,4.49) | 0.65 (0.17,2.47) | 0.25 (0.03,2.35) | 0.32 (0.07,1.40) |
| **Tro** | 5.30 (0.63,44.59) | 4.28 (0.96,18.96) | 3.42 (0.60,19.59) | 2.14 (0.34,13.38) | 1.53 (0.47,4.99) | 1.22 (0.28,5.31) | 1.03 (0.16,6.57) | 1.01 (0.15,6.94) | **Tro** | 0.94 (0.27,3.31) | 0.73 (0.17,3.06) | 0.65 (0.16,2.60) | 0.26 (0.04,1.74) | 0.32 (0.09,1.13) |
| **Ond** | 5.65 (0.59,53.78) | 4.56 (0.68,30.48) | 3.65 (0.47,28.20) | 2.28 (0.60,8.63) | 1.63 (0.45,5.96) | 1.31 (0.21,8.02) | 1.10 (0.13,9.34) | 1.07 (0.12,9.68) | 1.07 (0.30,3.77) | **Ond** | 0.78 (0.14,4.19) | 0.69 (0.12,3.96) | 0.27 (0.03,2.26) | 0.34 (0.06,1.79) |
| **Dro** | 7.27 (1.08,48.74) | 5.87 (0.97,35.50) | 4.69 (0.89,24.69) | 2.93 (0.34,25.08) | 2.10 (0.57,7.81) | 1.68 (0.42,6.65) | 1.42 (0.24,8.47) | 1.38 (0.22,8.58) | 1.37 (0.33,5.76) | 1.29 (0.24,6.93) | **Dro** | 0.89 (0.26,3.08) | 0.35 (0.10,1.26) | 0.44 (0.13,1.44) |
| **Dex** | 8.14 (0.93,71.03) | 6.57 (1.27,34.07) | 5.26 (1.55,17.79) | 3.29 (0.37,29.40) | 2.36 (0.52,10.69) | 1.88 (0.81,4.34) | 1.59 (0.37,6.77) | 1.55 (0.40,5.93) | 1.54 (0.38,6.15) | 1.44 (0.25,8.22) | 1.12 (0.33,3.86) | **Dex** | 0.39 (0.07,2.33) | 0.49 (0.26,0.93) |
| **Mid** | 20.70 (2.09,204.70) | 16.72 (1.84,151.83) | 13.37 (1.65,108.55) | 8.35 (0.69,101.45) | 5.99 (0.96,37.33) | 4.78 (0.73,31.24) | 4.05 (0.45,36.33) | 3.94 (0.42,36.54) | 3.91 (0.57,26.65) | 3.67 (0.44,30.31) | 2.85 (0.80,10.21) | 2.54 (0.43,15.04) | **Mid** | 1.24 (0.22,7.12) |
| **Pla** | 16.65 (1.99,139.58) | 13.44 (2.91,62.17) | 10.75 (3.17,36.46) | 6.72 (0.80,56.40) | 4.81 (1.15,20.21) | 3.85 (1.79,8.26) | 3.25 (0.84,12.57) | 3.17 (0.71,14.05) | 3.14 (0.88,11.17) | 2.95 (0.56,15.50) | 2.29 (0.70,7.53) | 2.04 (1.07,3.90) | 0.80 (0.14,4.60) | **Pla** |

Dark gray: Comparison, Light gray: Column compared with row, White: Row compared with column. Data are RRs (95% CI) in the column-defining treatment compared with the row-defining treatment or row-defining treatment compared with the column-defining treatment. For column compared with row, RRs higher than 1 favour the column-defining treatment. For row compared to column, RRs lower than 1 favour the row-defining treatment. RR = risk ratio. CI = confidence interval.

Pro: proprofol; Tro: tropisetron; Ram: ramosetron; Dex: dexamethasone; Pal: palonosetron; Met: metoclopramide; Gra: granisetron; Gin: oral ginger; Ond: ondansetron; Dro: droperidol; Mid: midazolam; Pla: placebo

**Table 5. League table for POV.**

| | | | | | | | |
|---|---|---|---|---|---|---|---|
| **Tro+Pro** | 0.26 (0.03,2.57) | 0.14 (0.03,0.78) | 0.10 (0.02,0.68) | 0.10 (0.02,0.62) | 0.08 (0.01,0.88) | 0.06 (0.01,0.40) | 0.03 (0.01,0.18) |
| 3.81 (0.39,37.32) | **Ram+Dex** | 0.54 (0.09,3.14) | 0.39 (0.08,1.90) | 0.37 (0.06,2.18) | 0.31 (0.04,2.62) | 0.24 (0.05,1.08) | 0.13 (0.03,0.60) |
| 7.05 (1.29,38.50) | 1.85 (0.32,10.73) | **Tro** | 0.73 (0.22,2.39) | 0.69 (0.25,1.91) | 0.57 (0.09,3.78) | 0.45 (0.15,1.36) | 0.24 (0.10,0.56) |
| 9.69 (1.46,64.12) | 2.54 (0.53,12.29) | 1.37 (0.42,4.51) | **Ram** | 0.94 (0.30,3.00) | 0.78 (0.13,4.81) | 0.62 (0.23,1.63) | 0.33 (0.14,0.78) |
| 10.28 (1.61,65.73) | 2.70 (0.46,15.87) | 1.46 (0.52,4.07) | 1.06 (0.33,3.38) | **Gra** | 0.83 (0.12,5.63) | 0.65 (0.21,2.06) | 0.35 (0.14,0.86) |
| 12.34 (1.14,134.13) | 3.24 (0.38,27.52) | 1.75 (0.26,11.61) | 1.27 (0.21,7.81) | 1.20 (0.18,8.11) | **Dex+Gin** | 0.79 (0.17,3.63) | 0.42 (0.08,2.25) |
| 15.71 (2.52,98.03) | 4.12 (0.92,18.40) | 2.23 (0.73,6.78) | 1.62 (0.61,4.29) | 1.53 (0.49,4.80) | 1.27 (0.28,5.87) | **Dex** | 0.53 (0.26,1.09) |
| 29.67 (5.50,160.05) | 7.79 (1.67,36.38) | 4.21 (1.80,9.86) | 3.06 (1.28,7.35) | 2.88 (1.16,7.16) | 2.40 (0.44,13.02) | 1.89 (0.92,3.87) | **Pla** |

Dark gray: Comparison, Light gray: Column compared with row, White: Row compared with column. Data are RRs (95% CI) in the column-defining treatment compared with the row-defining treatment or row-defining treatment compared with the column-defining treatment. For column compared with row, RRs higher than 1 favour the column-defining treatment. For row compared to column, RRs lower than 1 favour the row-defining treatment. RR = risk ratio. CI = confidence interval.

Pro: proprofol; Tro: tropisetron; Ram: ramosetron; Dex: dexamethasone; Gra: granisetron; Gin: oral ginger; Pla: placebo

The evaluation of the network inconsistency using the design-by-treatment interaction model suggested a significant network inconsistency [$F_{(8,13)}$ = 12.98; P = 0.1126].

There were 12 closed loops in the network generated from the comparisons of the use of rescue antiemetics, but 3 loops (Ond-Tro-Met [75], Pro-Dro-Met [71], and Tro-Dex-Tro +Dex [17]) consisted of only multi-arm trials. Although most loops showed no significance in the local inconsistency between the direct and indirect point estimates, the 5-6-7-10 loop (which included Gra-Tro-Dex-Ram) showed significant inconsistency (S-Fig 4D in S1 File).

Treatment with Tro+Pro, Ram+Dex, Ram, Tro, Tro+Dex, Gra, and Dex reduced the use of rescue antiemetics compared with Con in the overall phase only in terms of their 95% CIs, but not their 95% PrIs (S-Fig 5D and S-Fig 6D in S1 File).

The rankogram and cumulative ranking plot showed Tro+Pro to be the most effective pharmacologic intervention in reducing the use of rescue antiemetics (S-Fig 7D, S-Fig 8D in S1 File and **Table 6**).

The expected mean rankings and the SUCRA plots showed that the use of antiemetics was lowest in Tro+Pro (3.9%), followed by Ram+Dex (6.9%), and in Pro (25.1%) (**Fig 3D**).

The comparison-adjusted funnel plots suggested a less likely publication bias (S-Fig 9D in S1 File).

**Complete response.**   A total of four studies (556 patients) measured the frequencies of complete response (**Fig 2E1**).

One study, which compared the efficacy of Pal vs Pal+Dex, was excluded from the NMA because it was separated from the other loops [60]. Thus, six pharmacologic interventions (Tro +Pro, Tro+Dex, Tro, Dex, Pal, and Pal+Dex) were compared in three studies (472 patients) [8, 17, 73] (**Fig 2E2**).

**Table 6. League table for rescue anti-emetics.**

| | | | | | | | | | | | | | |
|---|---|---|---|---|---|---|---|---|---|---|---|---|---|
| **Tro+Pro** | 1.52 (0.20,11.76) | 4.05 (0.31,52.27) | 5.47 (0.98,30.50) | 6.98 (1.37,35.51) | 9.92 (1.71,57.66) | 10.26 (1.47,71.37) | 11.29 (2.15,59.31) | 12.06 (1.88,77.33) | 17.25 (1.76,168.75) | 17.25 (2.66,111.85) | 30.91 (2.73,350.51) | 30.05 (5.96,151.55) | 233.09 (4.79,11334.36) |
| 0.66 (0.09,5.09) | **Ram+Dex** | 2.66 (0.24,29.63) | 3.60 (0.98,13.18) | 4.60 (1.16,18.27) | 6.53 (1.49,28.57) | 6.75 (1.37,33.35) | 7.43 (2.15,25.71) | 7.94 (1.53,41.20) | 11.35 (1.37,93.83) | 11.35 (2.16,59.67) | 20.34 (2.10,197.04) | 19.78 (5.53,70.69) | 153.41 (3.47,6773.07) |
| 0.25 (0.02,3.19) | 0.38 (0.03,4.17) | **Pro** | 1.35 (0.16,11.49) | 1.72 (0.24,12.42) | 2.45 (0.29,20.88) | 2.53 (0.25,25.65) | 2.79 (0.35,22.44) | 2.98 (0.41,21.56) | 4.26 (0.74,24.42) | 4.26 (0.74,24.42) | 7.63 (1.10,52.87) | 7.42 (0.94,58.92) | 57.57 (3.10,1070.60) |
| 0.18 (0.03,1.02) | 0.28 (0.08,1.02) | 0.74 (0.09,6.29) | **Ram** | 1.28 (0.56,2.92) | 1.81 (0.72,4.59) | 1.87 (0.56,6.24) | 2.06 (1.07,3.99) | 2.20 (0.65,7.47) | 3.15 (0.52,19.07) | 3.15 (0.91,10.87) | 5.65 (0.78,41.06) | 5.49 (2.98,10.11) | 42.60 (1.14,1594.98) |
| 0.14 (0.03,0.73) | 0.22 (0.05,0.87) | 0.58 (0.08,4.18) | 0.78 (0.34,1.79) | **Tro** | 1.42 (0.62,3.27) | 1.47 (0.44,4.92) | 1.62 (0.83,3.17) | 1.73 (0.70,4.24) | 2.47 (0.50,12.22) | 2.47 (0.98,6.21) | 4.43 (0.73,26.86) | 4.30 (2.30,8.06) | 33.38 (0.98,1135.98) |
| 0.10 (0.02,0.59) | 0.15 (0.04,0.67) | 0.41 (0.05,3.48) | 0.55 (0.22,1.40) | 0.70 (0.31,1.62) | **Gra** | 1.03 (0.27,3.94) | 1.14 (0.47,2.75) | 1.22 (0.36,4.14) | 1.74 (0.29,10.54) | 1.74 (0.50,6.02) | 3.12 (0.43,22.69) | 3.03 (1.37,6.68) | 23.49 (0.63,880.62) |
| 0.10 (0.01,0.68) | 0.15 (0.03,0.73) | 0.39 (0.04,4.00) | 0.53 (0.16,1.78) | 0.68 (0.20,2.28) | 0.97 (0.25,3.69) | **Dex+Gin** | 1.10 (0.40,3.01) | 1.18 (0.26,5.30) | 1.68 (0.23,12.48) | 1.68 (0.37,7.69) | 3.01 (0.34,26.42) | 2.93 (0.97,8.84) | 22.73 (0.55,946.03) |
| 0.09 (0.02,0.47) | 0.13 (0.04,0.47) | 0.36 (0.04,2.89) | 0.48 (0.25,0.94) | 0.62 (0.32,1.21) | 0.88 (0.36,2.13) | 0.91 (0.33,2.48) | **Dex** | 1.07 (0.35,3.28) | 1.53 (0.27,8.66) | 1.53 (0.49,4.78) | 2.74 (0.40,18.76) | 2.66 (1.69,4.20) | 20.65 (0.57,748.92) |
| 0.08 (0.01,0.53) | 0.13 (0.02,0.65) | 0.34 (0.05,2.43) | 0.45 (0.13,1.54) | 0.58 (0.24,1.42) | 0.82 (0.24,2.80) | 0.85 (0.19,3.84) | 0.94 (0.30,2.87) | **Ond** | 1.43 (0.29,7.12) | 1.43 (0.56,3.63) | 2.56 (0.42,15.64) | 2.49 (0.83,7.45) | 19.33 (0.57,659.50) |
| 0.06 (0.01,0.57) | 0.09 (0.01,0.73) | 0.23 (0.04,1.35) | 0.32 (0.05,1.92) | 0.40 (0.08,2.00) | 0.58 (0.09,3.49) | 0.59 (0.08,4.41) | 0.65 (0.12,3.71) | 0.70 (0.14,3.48) | **Dro** | 1.00 (0.27,3.69) | 1.79 (0.78,4.13) | 1.74 (0.31,9.70) | 13.51 (0.45,406.81) |
| 0.06 (0.01,0.38) | 0.09 (0.02,0.46) | 0.23 (0.04,1.35) | 0.32 (0.09,1.09) | 0.40 (0.16,1.02) | 0.58 (0.17,1.99) | 0.59 (0.13,2.72) | 0.65 (0.21,2.05) | 0.70 (0.28,1.78) | 1.00 (0.27,3.69) | **Met** | 1.79 (0.38,8.44) | 1.74 (0.57,5.31) | 13.51 (0.45,406.82) |
| 0.03 (0.00,0.37) | 0.05 (0.01,0.48) | 0.13 (0.02,0.91) | 0.18 (0.02,1.29) | 0.23 (0.04,1.37) | 0.32 (0.04,2.34) | 0.33 (0.04,2.91) | 0.37 (0.05,2.50) | 0.39 (0.06,2.38) | 0.56 (0.24,1.29) | 0.56 (0.12,2.63) | **Mid** | 0.97 (0.14,6.56) | 7.54 (0.23,251.09) |
| 0.03 (0.01,0.17) | 0.05 (0.01,0.18) | 0.13 (0.02,1.07) | 0.18 (0.10,0.34) | 0.23 (0.12,0.44) | 0.33 (0.15,0.73) | 0.34 (0.11,1.03) | 0.38 (0.24,0.59) | 0.40 (0.13,1.20) | 0.57 (0.10,3.20) | 0.57 (0.19,1.75) | 1.03 (0.15,6.94) | **Pla** | 7.76 (0.22,278.98) |
| 0.00 (0.00,0.21) | 0.01 (0.00,0.29) | 0.02 (0.00,0.32) | 0.02 (0.00,0.88) | 0.03 (0.00,1.02) | 0.04 (0.00,1.60) | 0.04 (0.00,1.83) | 0.05 (0.00,1.76) | 0.05 (0.00,1.77) | 0.07 (0.00,2.23) | 0.07 (0.00,2.23) | 0.13 (0.00,4.42) | 0.13 (0.00,4.64) | **Int** |

Dark gray: Comparison, Light gray: Column compared with row, White: Row compared with column. Data are RRs (95% CI) in the column-defining treatment compared with the row-defining treatment or row-defining treatment compared with the column-defining treatment. For column compared with row, RRs higher than 1 favour the column-defining treatment. For row compared to column, RRs lower than 1 favour the row-defining treatment. RR = risk ratio. CI = confidence interval.

Pro: proprofol; Tro: tropisetron; Ram: ramosetron; Dex: dexamethasone; Gra: granisetron; Gin: oral ginger; Ond: ondansetron; Dro: droperidol; Met: metoclopramide; Md: midazolam; Pla: placebo; Int: Intralipid

The evaluation of the network inconsistency using the design-by-treatment interaction model suggested that there was a significant inconsistency [F(1,2) = 0.92; P = 0.9038].

There were three closed loops in the network generated from the comparisons of the complete response; however, two (Pla-Tro-Tro+Pro [8] and Tro-Dex-Tro+Dex [17]) consisted of only multi-arm trials. Although most loops showed no significance in the local inconsistency between the direct and indirect point estimates, the 5-6-7-10 loop (which included Gra-Tro-Dex-Ram) showed significant inconsistency (S-Fig 4E in S1 File).

There was no significance in the local inconsistency between the direct and indirect point estimates (S-Fig 5E in S1 File).

Tro, Dex, Tro+Pro, and Tro+Dex showed higher complete responses than Pla in terms of the 95% CIs. Tro+Pro had a higher complete response than Tro and Dex. Tro+Dex also showed a higher complete response than Dex (S-Fig 6E in S1 File).

The rankogram and cumulative ranking plot showed that Tro+Pro had the highest complete response in the overall phase (S-Fig 7E in S1 File).

The cumulative ranking plot was drawn and the SUCRA probabilities of the different pharmacologic interventions for the complete response in the overall phase were calculated (S-Fig 8E in S1 File and **Table 7**).

The expected mean rankings and SUCRA values of each airway device are presented in **Fig 3E**.

The complete response was highest with Tro+Pro (96.6%), followed by Tro+Dex (75.7%), Tro (48.8%). The comparison-adjusted funnel plots show that they were symmetrical around the zero line, which suggests limited publication bias (S-Fig 9E in S1 File).

## Safety

The extracted data for safety issues were presented in **S2 Table**. As a lot of studies did not report the outcomes on safety issues, network meta-analysis was not performed.

## Quality of evidence

Three outcomes were evaluated using the **Grading** of Recommendations Assessment, Development and **Evaluation** (GRADE) system. The evidence quality for each outcome was low or moderate (**Table 8**). All the quality of pooled analysis showed moderate except that in complete response which shows low.

**Table 7. League table for complete response.**

| Tro+Pro | 2.05 | 3.69 | 5.33 | 15.90 |
|---|---|---|---|---|
|  | (0.55,7.66) | (1.27,10.70) | (1.67,16.99) | (5.26,48.13) |
| 0.49 | Tro+Dex | 1.80 | 2.60 | 7.76 |
| (0.13,1.82) |  | (0.76,4.25) | (1.11,6.07) | (2.87,20.98) |
| 0.27 | 0.56 | Tro | 1.44 | 4.31 |
| (0.09,0.79) | (0.24,1.31) |  | (0.73,2.86) | (2.02,9.17) |
| 0.19 | 0.38 | 0.69 | Dex | 2.98 |
| (0.06,0.60) | (0.16,0.90) | (0.35,1.37) |  | (1.56,5.72) |
| 0.06 | 0.13 | 0.23 | 0.34 | Pla |
| (0.02,0.19) | (0.05,0.35) | (0.11,0.49) | (0.17,0.64) |  |

Dark gray: Comparison, Light gray: Column compared with row, White: Row compared with column. Pro: proprofol; Tro: tropisetron; Dex: dexamethasone; Pla: placeb

**Table 8. The GRADE evidence quality for each outcome.**

| Outcomes | Number of studies | Quality assessment | | | | | Quality |
|---|---|---|---|---|---|---|---|
| | | Risk of bias | Inconsistency | Indirectness | Imprecision | Publication bias | |
| **PONV** | 10 | serious | not serious | not serious | not serious | not serious | ⊕⊕⊕◯ Moderate |
| **PON** | 13 | serious | not serious | not serious | not serious | not serious | ⊕⊕⊕◯ Moderate |
| **POV** | 9 | serious | not serious | not serious | not serious | not serious | ⊕⊕⊕◯ Moderate |
| **Use of rescue antiemetics** | 17 | serious | not serious | not serious | not serious | not serious | ⊕⊕⊕◯ Moderate |
| **Complete response** | 3 | serious | not serious | not serious | serious | not serious | ⊕⊕◯◯ Low |

PON; postoperative nausea, POV; postoperative vomiting, PONV; postoperative nausea and vomiting

## Discussion

This NMA demonstrated that propofol and tropisetron, alone and in combination, and ramosetron in combination with dexamethasone were superior in 1) reducing the incidence of PONV, PON and POV; 2) reducing the use of rescue antiemetics, and 3) enhancing complete response compared to the other pharmacologic interventions. In our NMA, propofol was the most effective pharmacologic intervention as a strategy for preventing PON and PONV and the third most effective pharmacologic intervention for reducing use of rescue antiemetics in the overall phase. Tropisetron was efficacious in reducing POV, PONV, and in enhancing the complete response. Tropisetron combined with propofol was the most effective pharmacologic intervention in preventing POV, in reducing the use of rescue antiemetics, and in enhancing complete response. Lastly, ramosetron combined with dexamethasone was also effective in preventing PON and POV, and in reducing the use of rescue antiemetics.

Propofol-based anesthesia is known to decrease the incidence of PONV compared with volatile anesthetics [26, 76]. Its efficacy has been demonstrated when administered for both induction and maintenance anesthesia, but not when given as a bolus dose before the end of surgery for preventing PONV. In our NMA, propofol, given as a bolus before the end of surgery, was the most effective treatment regimen in preventing PON and PONV. These results are supported by a previous report which demonstrated that propofol was efficacious in treating PONV at plasma concentrations that do not produce increased sedation [77]. It is also reported that propofol given for elective cesarean section under spinal anesthesia at sub-hypnotic doses decreased the incidence of PONV without unwanted sedative and respiratory or cardiovascular side effects [78]. Although the exact mechanism by which propofol prevents emesis is unknown, antagonism of the dopaminergic [26] and serotonergic pathways, modulation of the subcortical pathways [79], and direct depressant effect on the chemoreceptor trigger zone, the vagal nuclei, and other centers [80] were suggested as possible antiemetic mechanisms.

Many chemoreceptors and associated pathways are involved in the mechanism of PONV; various antiemetics, including 5-HT$_3$ receptor antagonists, glucocorticoids, anticholinergics, neurokinin-1 receptor antagonists, dopamine receptor antagonists, cannabinoids, and antihistamines are used in clinical practice. Of these, 5-HT$_3$ receptor antagonists have been well-documented to be effective in preventing and treating PONV and are frequently prescribed clinically. In our NMA, tropisetron was highly efficacious in reducing POV, PONV, and in

enhancing the complete response, while ramosetron in combination with dexamethasone was effective in the prevention of PON, POV, and in reducing the use of rescue antiemetics.

Tropisetron is a highly potent and selective 5-HT$_3$ receptor antagonist [81], and the findings in our NMA is supported by those reported by a previous meta-analysis [82], as well as RCTs [83, 84], which showed that tropisetron was effective and well-tolerated in the prevention or treatment of PONV in other types of surgery.

As multifactorial etiologies of PONV have been identified, and none of the currently available antiemetics are capable of completely eliminating the risk of PONV, it seems logical to use a combination of antiemetics with different mechanisms of action. In our NMA, a combination of antiemetics with different mechanisms of action was highly effective in preventing PONV. Ramosetron combined with dexamethasone showed good efficacy in preventing PON, POV, and in reducing the use of antiemetics. Tropisetron combined with propofol was efficacious in preventing POV; the combination reduced the need for rescue antiemetics and enhanced complete response. The combination of tropisetron with dexamethasone also enhanced complete response. These findings are supported by previous studies, which demonstrated combined antiemetic therapy to be more effective than monotherapy. For instance, tropisetron combined with propofol infusion was more effective than tropisetron alone [85], and the combination of ramosetron and dexamethasone was more effective than ramosetron alone for preventing PONV in patients undergoing thyroid surgery [15]. Furthermore, the tropisetron-propofol combination decreased the frequency of PONV to as low as 17% in patients undergoing thyroidectomy [8].

The NMA performed in this study has some limitations. First, overall and local inconsistency was suggested in some outcomes. Although we validated the transitivity assumptions by examining the comparability of patient eligibility criteria, demographics and types of pharmacological interventions, study design, the risk of bias as a potential treatment-effect modifier across comparisons before performing NMA, the risk of methodological heterogeneity, all of which were not considered, still exists. Second, only a limited number of studies were included, and the dose spectrums of the injected pharmacological interventions were wide. Moreover, two studies that compared Dro vs. Mid [67] and Pro vs. Dro vs. Met [71] for POV, and one study that compared Pal vs Pal+Dex [60] for complete response were separated from the loops and could not be compared with other drugs; hence, the collected data for such drugs were excluded in this NMA. Therefore, well-designed, large-scale RCTs that compare various antiemetic drugs, for which comparison was not performed in previous studies, should be conducted in future to validate the outcomes of our study. Lastly, this systematic review and NMA only focused to the results from thyroidectomy; therefore, the results cannot be extrapolated to patients receiving other type of surgery.

Despite the abovementioned limitations, our systematic review and NMA represent a fair evaluation of pharmacologic interventions used for reducing PONV in patients undergoing thyroidectomy. The methodologies applied may be useful to other researchers aiming to conduct similar reviews. Furthermore, our NMA provides clinical evidence-based guidance to aid physicians in selecting an effective pharmacological intervention to prevent PONV after thyroidectomy.

## Conclusion

In conclusion, propofol, tropisetron, their combination, and ramosetron combined with dexamethasone was effective in preventing PON, POV, PONV, reducing the need for rescue antiemetics, and in enhancing complete response. However, considering the substantial heterogeneity and limited number of studies included, the results of our meta-analysis should be interpreted with caution.

## Supporting information

**S1 Checklist. PRISMA-NMA checklist.**
(DOCX)

**S1 Search Term.**
(DOCX)

**S1 File.**
(DOCX)

**S2 File.**
(DOCX)

**S1 Table.**
(DOCX)

**S2 Table.**
(DOCX)

**S3 Table.**
(DOCX)

**S1 Fig.**
(TIF)

**S2 Fig.**
(TIF)

**S3 Fig.**
(TIF)

**S4 Fig.**
(TIF)

**S5 Fig.**
(TIF)

**S6 Fig.**
(TIF)

**S7 Fig.**
(TIF)

**S8 Fig.**
(TIF)

**S9 Fig.**
(TIF)

**S10 Fig.**
(TIF)

**S11 Fig.**
(TIF)

**S12 Fig.**
(TIF)

**S13 Fig.**
(TIF)

**S14 Fig.**
(TIF)

## Acknowledgments

The difference between this article and the protocol is that Ahn EJ has joined as an investigator in this NMA in part of Data extraction and Study quality assessment.

## Author Contributions

**Conceptualization:** Ye Jin Cho, Geun Joo Choi, Eun Jin Ahn, Hyun Kang.

**Data curation:** Ye Jin Cho, Geun Joo Choi, Eun Jin Ahn, Hyun Kang.

**Formal analysis:** Hyun Kang.

**Funding acquisition:** Hyun Kang.

**Investigation:** Ye Jin Cho, Eun Jin Ahn, Hyun Kang.

**Methodology:** Geun Joo Choi, Eun Jin Ahn, Hyun Kang.

**Project administration:** Hyun Kang.

**Resources:** Eun Jin Ahn.

**Supervision:** Hyun Kang.

**Validation:** Geun Joo Choi, Hyun Kang.

**Visualization:** Ye Jin Cho, Hyun Kang.

**Writing – original draft:** Ye Jin Cho, Eun Jin Ahn, Hyun Kang.

**Writing – review & editing:** Geun Joo Choi, Hyun Kang.

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
