## [Decision Letter · Decision Letter 0]

11 Sep 2020

PONE-D-20-20400

Pharmacologic interventions for postoperative nausea and vomiting after thyroidectomy: a systematic review and network meta-analysis

PLOS ONE

Dear Dr. Kang,

Thank you for submitting your manuscript to PLOS ONE. After careful consideration, we feel that it has merit but does not fully meet PLOS ONE’s publication criteria as it currently stands. Therefore, we invite you to submit a revised version of the manuscript that addresses the points raised during the review process.

ACADEMIC EDITOR:

Your manuscript has been reviewed by two experts in the field, and they have found some points that need to be addressed before this manuscript is considered for publication. Please go through the reviewers' comments and consider addressing these points, and prepare a revised version.

In addition to provide a point-by-point response to the reviewers comments, please consider the following major editor's comments, that need to be addressed:

1. It is not clear how data extraction phase disagreements between both reviewers were handled.

2. The transitivity assessment can be improved. Was this performed for the whole network or per loop? Also, how was"comparability" defined. Was this just a visual, preliminary analysis of how they were similar? Or were some statistics used? How can we define "comparability" according to, for instance, to demographics?

3. Page 11, The description of the use of Inconsistency global test, the design-by-treatment interaction model, should have an appropriate citation.

4. You have chosen to present the PrIs. This is fine, but this intervals may ring more confusion to readers (mostly to clinicians). Since PrI is not a method that is used in all meta-analysis, it is important to provide more explanation in the methods section, of its use and its interpretation, if Fig. 6, and rankograms can be removed or if preferred presented as Supporting Information. This applies to all the outcomes

5. There are many figures that are not essential and one major output is not presented. I suggest reestructuring your figures and do the follwoing:

- present a league table at least for the primary outcome(s)

- Remove from the manuscript and move them all to supplemental file, the following: Rankograms, funnel plots and Inconsistency analyses. They are not key in the main document.

- Keep in the main manuscript: Network plots (consider putting all the Outcomes network plots in one single figure), League table and perhaps SUCRAS plot that has colors.

- League table deserves special mention. You have not provided league table and this is key as it provides all the results of all the possible comparisons. You can choose a League table that summaries one outcome or you can create a table with two outcomes. For more information check this paper by: Rouse et al. 2017 (https://www.ncbi.nlm.nih.gov/pmc/articles/PMC5247317/)

6. Finally, but not least, you have wrongly chosen to follow the PRISMA checklist. For NMAs you need to follow the specific guidance on NMAs: PRISMA-NMA (Hutton et al 2015)

We look forward to receiving your revised manuscript.

Kind regards,

Ivan D. Florez

Academic Editor

PLOS ONE

Journal Requirements:

2. Please ensure you have included the full electronic search strategy for at least one database and uploaded it as an additional file.

3. Please upload a new copy of Figure 5, S-Figure-A, S-Figure-B, S-Figure-C, S-Figure-M, S-Figure-L, S-Figure-H, S-Figure-G as the detail is not clear. Please follow the link for more information: https://blogs.plos.org/plos/2019/06/looking-good-tips-for-creating-your-plos-figures-graphics/" https://blogs.plos.org/plos/2019/06/looking-good-tips-for-creating-your-plos-figures-graphics/

Additional Editor Comments (if provided):

See above in specific Academic Editor comments.

Reviewers' comments:

Reviewer's Responses to Questions

**Comments to the Author**

1. Is the manuscript technically sound, and do the data support the conclusions?

Reviewer #1: Partly

Reviewer #2: Yes

2. Has the statistical analysis been performed appropriately and rigorously? 

Reviewer #1: I Don't Know

Reviewer #2: Yes

3. Have the authors made all data underlying the findings in their manuscript fully available?

Reviewer #1: Yes

Reviewer #2: Yes

4. Is the manuscript presented in an intelligible fashion and written in standard English?

Reviewer #1: Yes

Reviewer #2: No

5. Review Comments to the Author

Reviewer #1: Pharmacologic interventions for postoperative nausea and vomiting after thyroidectomy: a systematic review and network meta-analysis

I appreciate the opportunity to read and evaluate this systematic review and NMA. Certainly, its purpose is highly important in perioperative care. It is a remarkable effort to synthesize all evidence regarding a specific condition and on this specific population. The manuscript is well organized and written clearly enough. As the research question is interesting and has future implications, the manuscript can be considered, but not without a major revision addressing and solving the discussed points, some of them critical.

Major concerns.

- Restriction of the population to only thyroidectomy.

- Absence of analysis of at least one adverse related event. Any NMA should consider principal outcomes (efficacy), as well as, safety/secondary/adverse-events outcomes as equally important. Moreover, why those outcomes were clearly stated in the published protocol but not reported in this final report.

- Not considering potential correlation in multi-arm trials.

- GRADE assessment should consider at least one safety-related outcome.

Detailed comments.

Introduction.

- Introduction is clear and sound. I really feel that the increase on incidence of PONV after thyroidectomy is very important but what about other types of even high-incidence of PONV. You did a very important effort synthetizing the overall evidence of pharmacological interventions, and any reader may ask if your conclusions are really valid to other surgeries?

- Underlying mechanism of PONV in thyroidectomy are so different from other surgeries? enough to isolate and study thyroidectomy alone? I think you should add some related point s to your introductions and probably to your discussion. In line with this comment, there is one Cochrane protocol ongoing including all surgeries under general anesthesia (https://www.cochranelibrary.com/cdsr/doi/10.1002/14651858.CD012859/full/es).

- While it is understandable you choose only thyroidectomy in order to study PONV interventions I recommend to present that as a limitation. Please state stronger and put more emphasis on why do you consider this NMA only for thyroidectomy.

- Aim is very clear and focused on prevention.

Methods.

- Good to know that this systematic review has followed in detail PRISMA-ext guidelines and registered/published their protocol. However, you should clarify the contribution of each author considering you add one from the protocol to this report.

- Please add the precise dates of search on each database.

- While we know that incidence of PONV is higher in early postop periods and important outcomes for patients are long-lasting ones, why did you choose the earliest time-point in each constructed period? Why not the last? Any major reason or argument?

- Please provide any effort to run a grey literature search. Did you use any?

- It is quite interesting you did not report (as outcomes) any adverse, secondary or safety related outcomes for any intervention? Are they so safe? Or did you put much more interest in efficacy and not in safety. Please clarify in detail and considering as strong limitation of your report considering many trials do report safety outcomes. Let consider some potential safety/serious adverse events from those interventions: Arrhythmia, QT prolongation, Extrapyramidal symptoms, Postoperative wound infection, Sedation/drowsiness, among others.

- Please explain this clear reporting bias: Safety outcomes were clearly stated in the published protocol but not reported in this final report (i.e. The severity of PON, POV, and PONV; the use of rescue antiemetics; the incidence of complete response; and safety issues, such as headache, dizziness, drowsiness, and constipation, will be also assessed.) taken from: (https://www.ncbi.nlm.nih.gov/pmc/articles/PMC6407968/)

- Regarding transitivity, how did you deal with time of administrations of preventive pharmacological interventions. Probably most of them were preoperatively. Please add some information about the analysis of methodological approach to different doses (low/high doses), time of administration of preventive drugs.

- Considering preoperative individual and variable risk of each included individuals, any major reason not to consider a Bayesian approach to NMA? You consider such a homogeneous population and may be that is the reason?

- You included several multi-arm trials. Comparisons within multi‐arm studies are correlated, did you adjust the standard error of each two‐arm comparison to multi‐arm studies. Please consider the method of Rücker and Schwarzer (back‐calculated standard errors in the weighted least‐square estimator).

- Considering combined treatments (e.g. A + B, A + B + C, B + C) are additive sums of their components, why do not present analysis to all mono‐prophylaxis and later to combination prophylaxis? Please argue about your analysis.

Results.

- In line 261, I believe you are talking about “preoperative seventeen…”. That should be more clear from methods section. You are considering preventing drugs pre or intraop. administered but not postoperative. Please clarify this in detail.

- Not very sure if “Oral Gin” should be considered a pharmacological active agent. May be need some short clarification in discussion.

- Fig 2a is very clear and describe very well the overall evidence of this population. However, the main node “Con” is not previously described as an intervention. It is referring to Placebo, please clarify. Also for the text about loops and direct/indirect evidence.

- Considering reference treatment to Con (Placebo) is quite worrying. Did you consider to use any other widely used intervention as reference? let say Dexamethasone?

Discussion.

- You clearly state that “Its efficacy has been demonstrated when administered for both induction and maintenance anesthesia, but not when given as a bolus dose before the end of surgery for preventing PONV”. However, the overall research process does not clearly address the timing or doses of administered interventions on each trial. Please see above in previous comments.

- Again, “propofol was efficacious 615 in treating PONV at plasma concentrations that do not produce increased sedation”. Did you consider any AR, Serious AE or any safety related outcomes?

- While limitations are clear, why not to suggest exactly what types of comparisons are more needed at large-scale RCT? That should be an interesting finding for further RCTs.

- Please explain the reporting bias regarding safety outcomes.

Reviewer #2: This network meta-analysis is to determine the effectiveness of pharmacologic interventions for preventing postoperative nausea and vomiting (PONV) in patients undergoing thyroidectomy. The primary endpoints were the incidences of postoperative nausea (PON), postoperative vomiting (POV), PONV, use of rescue antiemetic and incidence of complete response in the overall postoperative phases. The secondary endpoints were the same parameters assessed in the early, middle, and late postoperative phases. The surface under the cumulative ranking curve (SUCRA) values and rankograms were used to present the hierarchy of pharmacologic interventions.

Twenty-nine studies (n=3,755 patients) that investigated 18 different pharmacologic interventions were included. The incidence of PONV among the overall postoperative phases was lowest with propofol alone (16.1%). The least usage of rescue antiemetics among the overall postoperative phases and the highest complete response was observed with tropisetron and propofol combination (3.9% and 96.6%, respectively). The authors concluded that propofol and tropisetron alone and in combination, and the ramosetron and dexamethasone combination effectively prevented PONV in patients undergoing thyroidectomy, with some heterogeneity observed in this NMA of full-text reports.

This is a detailed network meta-analysis . However, it has several primary outcomes, PON, PON and PONV. Also, there are 18 different pharmacologic interventions. Thus, there are many pages of results, large number of figures, and many tables making this difficult to read and understand the true impact of this network meta-analysis. It will benefit in reduction of the primary end points. As the author pointed out in their objective, the study is to determine the effectiveness of pharmacologic interventions for preventing postoperative nausea and vomiting (PONV) in patients undergoing thyroidectomy. The primary end point is PONV. The authors have chosen to present PON first, then POV, then PONV results. Suggest that the authors presented the PONV data first. For the data of PON and the data on POV, the authors could highlight the key differences in results without going into details. Also some of the tables and some of the figures can be inserted into supplementary materials.

Ref 66: dexamethasone (Dex)+Oral ginger. The study indicated that it is pharmacological intervention. Does oral ginger consider pharmacological?

The postoperative period was divided into the early, middle, late, and overall phases. The

early phase was defined as 0–6 h postoperatively; middle phase, 6–24 h postoperatively; and

late phase, more than 24 h postoperatively. If a study reported data at multiple time points

within the same phase, data from the first time point were selected as the outcome of interest

(e.g., if the study reported data at 0 h, 2 h, 4 h, and 6 h postoperatively, we only included the

data at 0 h as the early phase). This classification may have biased the results towards propofol, since a bolus given at end of surgery may abort PONV immediately, but it may not have sustained effect except if given as infusion.

To ensure the inclusion of maximum number of studies, any PON, POV, and PONV data from studies that do not mention a specific time point were defined as the overall phase. What is meant by overall phase, is this 6 to 24h? This may also misclassify groups.

6. PLOS authors have the option to publish the peer review history of their article (what does this mean?). If published, this will include your full peer review and any attached files.

Reviewer #1: **Yes: **Jose Andres Calvache

Reviewer #2: No

---

## [Author Response · Author response to Decision Letter 0]

24 Oct 2020

23th October 18, 2020

Dear Dr Ivan D. Florez, Academic Editor of ‘Plos One’,

Re: Manuscript number PONE-D-20-20400, “Pharmacologic interventions for postoperative nausea and vomiting after thyroidectomy: a systematic review and network meta-analysis”

We thank sincerely the academic editor and reviewers of the ‘Plos One’ for taking their precious time to review our paper. Your constructive, meticulous and considerate comments were great guidance our aforementioned manuscript. According to your precious comments and suggestions, we sincerely and earnestly tried to response for your comments. We want to express my heartfelt gratitude for your comments once more. 

We have made some corrections in the manuscript after going over your comments. We highlighted the modification made to the original document by using red colored text. The changes are summarized below:

Academic editor’s comment

Your manuscript has been reviewed by two experts in the field, and they have found some points that need to be addressed before this manuscript is considered for publication. Please go through the reviewers' comments and consider addressing these points, and prepare a revised version.

In addition to provide a point-by-point response to the reviewers comments, please consider the following major editor's comments, that need to be addressed:

1. It is not clear how data extraction phase disagreements between both reviewers were handled.

Our response: Thank you for editor’s suggestions for the betterment of the manuscript. According to editor’s suggestion, we clarified the data extraction phase disagreements in the manuscript. (from line 185 to line 186 on page 10 in the Methods section)

2. The transitivity assessment can be improved. Was this performed for the whole network or per loop? Also, how was "comparability" defined. Was this just a visual, preliminary analysis of how they were similar? Or were some statistics used? How can we define "comparability" according to, for instance, to demographics?

Our response: Thank you for editor’s considerate comments and suggestions for the betterment of the manuscript. According to your comment, we revised the description on transitivity assumption. (from line 210 to line 216 on page 11 in the Methods section).

Our response: According to editor’s comment, we added the citation for it. (line 225 on page 12 in the Methods section).

4. You have chosen to present the PrIs. This is fine, but this interval may ring more confusion to readers (mostly to clinicians). Since PrI is not a method that is used in all meta-analysis, it is important to provide more explanation in the methods section, of its use and its interpretation, if Fig. 6, and rankograms can be removed or if preferred presented as Supporting Information. This applies to all the outcomes

 Our response: Thank you for editor’s considerate comments and suggestions for the betterment of the manuscript. According to your comment, we described about predictive interval. (from line 233 to line 238 on page 12 in the Methods section). 

And we moved the Fig. 6, and rankograms to Supporting Information.

5. There are many figures that are not essential and one major output is not presented. I suggest restructuring your figures and do the following:

- present a league table at least for the primary outcome(s)

- Remove from the manuscript and move them all to supplemental file, the following: Rankograms, funnel plots and Inconsistency analyses. They are not key in the main document. 

- Keep in the main manuscript: Network plots (consider putting all the Outcomes network plots in one single figure), League table and perhaps SUCRAS plot that has colors.

- League table deserves special mention. You have not provided league table and this is key as it provides all the results of all the possible comparisons. You can choose a League table that summaries one outcome or you can create a table with two outcomes. For more information check this paper by: Rouse et al. 2017 (https://www.ncbi.nlm.nih.gov/pmc/articles/PMC5247317/)

 Our response: According to reviewer’s comment, we restructured our figures and provide the league table for all outcomes.

6. Finally, but not least, you have wrongly chosen to follow the PRISMA checklist. For NMAs you need to follow the specific guidance on NMAs: PRISMA-NMA (Hutton et al 2015)

 Our response: Thank you for your considerate comments and suggestions. According to reviewer’s comment, we provide the checklist of PRISMA-NMA

Reviewers' comments:

Reviewer #1: Pharmacologic interventions for postoperative nausea and vomiting after thyroidectomy: a systematic review and network meta-analysis

I appreciate the opportunity to read and evaluate this systematic review and NMA. Certainly, its purpose is highly important in perioperative care. It is a remarkable effort to synthesize all evidence regarding a specific condition and on this specific population. The manuscript is well organized and written clearly enough. As the research question is interesting and has future implications, the manuscript can be considered, but not without a major revision addressing and solving the discussed points, some of them critical.

Major concerns.

- Restriction of the population to only thyroidectomy.

Our response: Thank you for reviewer’s comment for the betterment of manuscript. According to reviewer’s comment, we added the description on PONV after thyroidectomy in the manuscript. (from line 59 to line 62 on page 4 in the Introduction section, from line 572 to line 574 on page 51 in the Discussion section). 

- Absence of analysis of at least one adverse related event. Any NMA should consider principal outcomes (efficacy), as well as, safety/secondary/adverse-events outcomes as equally important. Moreover, why those outcomes were clearly stated in the published protocol but not reported in this final report.

Our response: Thank you for your considerate comments and suggestions for the betterment of the manuscript. Although we planned analysis on safety issues, we did not report the results on safety issues, because almost studies did not report safety outcomes. However, as we agree with reviewer’s opinion, we provide the table on safety issues (Supplementary Table 2). Thank you again.

- Not considering potential correlation in multi-arm trials.

Our response: We sincerely appreciate your considerate comments and suggestions for the betterment of our manuscript. According to reviewer’s comment, we read the article by Rücker and Schwarzer, and learned a lot. We agree with your opinion that we should consider potential correlation in multi-arm trials and unit‐of‐analysis error issues. However, as you know well, we developed the protocol for this systematic review and network meta-analysis before the beginning of our work, which was registered to the PROSPERO network and also published in a peer reviewed journal. The purpose of these processes was to make the research process transparent and reproducible. In the protocol and published article, we did not consider potential correlation in multi-arm trials. Thus, although we totally agree with your opinion, we decide not to use the method of Rücker and Schwarzer. We shall apply this method what you guided this time in our future work. Again, thank you very much for your constructive comments. 

- GRADE assessment should consider at least one safety-related outcome.

Our response: As we made a comment above, we provide the table on safety tissues. However, as almost studies did not report safety issues and did not provide sufficient information, we cannot perform network meta-analysis. Thus, we decided not to make GRADA assessment. Thank you for your meticulous review.

Detailed comments.

Introduction.

- Introduction is clear and sound. I really feel that the increase on incidence of PONV after thyroidectomy is very important but what about other types of even high-incidence of PONV. You did a very important effort synthetizing the overall evidence of pharmacological interventions, and any reader may ask if your conclusions are really valid to other surgeries?

- Underlying mechanism of PONV in thyroidectomy are so different from other surgeries? enough to isolate and study thyroidectomy alone? I think you should add some related point s to your introductions and probably to your discussion. In line with this comment, there is one Cochrane protocol ongoing including all surgeries under general anesthesia (https://www.cochranelibrary.com/cdsr/doi/10.1002/14651858.CD012859/full/es).

- While it is understandable you choose only thyroidectomy in order to study PONV interventions I recommend to present that as a limitation. Please state stronger and put more emphasis on why do you consider this NMA only for thyroidectomy.

Our response: Thank you for reviewer’s comment for the betterment of manuscript. According to reviewer’s comment, we added the description on PONV after thyroidectomy in the manuscript. (from line 59 to line 62 on page 4 in the Introduction section, from line 572 to line 574 on page 51 in the Discussion section). 

- Aim is very clear and focused on prevention.

Our response: Thank you for reviewer’s comment.

Methods.

- Good to know that this systematic review has followed in detail PRISMA-ext guidelines and registered/published their protocol. However, you should clarify the contribution of each author considering you add one from the protocol to this report.

Our response: Thank you for your comment. As you can see, there were too many works to do than expected when we planned this NMA. Thus, we invited one investigator to join our NMA, and she had done data extraction and study quality assessment. We described the difference between this article and protocol in Acknowledgement section. (From line 590 to line 591 on page 52 in Acknowledgements section)

- Please add the precise dates of search on each database.

Our response: According to reviewer’s comment, we added the précised dates. (line 144 on page 8 in the Materials and Methods section).

- While we know that incidence of PONV is higher in early postop periods and important outcomes for patients are long-lasting ones, why did you choose the earliest time-point in each constructed period? Why not the last? Any major reason or argument?

Our response: Thank you for reviewer’s comment and suggestion. At the planning stage, we found that there were multiple time point issues. Thus, we made a search and discussed about that issue. Search showed that almost meta-analysis or network meta-analysis chose the earliest time point. (Ayako Yokoi, Takahiro Mihara, Koui Ka, et al. PloS One 2017; 12(10):e0186006, Kamal Awad, Hussien Ahmed, Abdelrahman Ibrahim Abushouk, et al. International Journal of Surgery 36 (2016) 152e163; Ahn EJ, Kang H, Choi GJ, et al. Anesth Analg 2016 Mar;122(3):664-76; Ahn EJ, Choi GJ, Kang H, et al. PLoS One 2016 Dec 19;11(12):e0168509) And, as you know, the incidence of PONV is higher in early time points. Thus, we decided to choose the earliest time point, and clarify the time-point at the protocol and published articles to make the process of study transparent. 

- Please provide any effort to run a grey literature search. Did you use any?

Our response: Thank you for your comments for the betterment of the manuscript. As you know, we developed the protocol for this systematic review and network meta-analysis before the beginning of our work. And at that stage, we did not plan to search grey literature. Thus, we did not search grey literature.

- It is quite interesting you did not report (as outcomes) any adverse, secondary or safety related outcomes for any intervention? Are they so safe? Or did you put much more interest in efficacy and not in safety. Please clarify in detail and considering as strong limitation of your report considering many trials do report safety outcomes. Let consider some potential safety/serious adverse events from those interventions: Arrhythmia, QT prolongation, Extrapyramidal symptoms, Postoperative wound infection, Sedation/drowsiness, among others.

- Please explain this clear reporting bias: Safety outcomes were clearly stated in the published protocol but not reported in this final report (i.e. The severity of PON, POV, and PONV; the use of rescue antiemetics; the incidence of complete response; and safety issues, such as headache, dizziness, drowsiness, and constipation, will be also assessed.) taken from: (https://www.ncbi.nlm.nih.gov/pmc/articles/PMC6407968/)

- GRADE assessment should consider at least one safety-related outcome.

Our response: As we made a comment above, we provide the table on safety issues. (Supplementary Table 2). However, as almost studies did not report safety issues and did not provide sufficient information, we cannot perform network meta-analysis. Thus, we decided not to make GRADA assessment. Thank you for your meticulous review.

- Regarding transitivity, how did you deal with time of administrations of preventive pharmacological interventions. Probably most of them were preoperatively. Please add some information about the analysis of methodological approach to different doses (low/high doses), time of administration of preventive drugs.

Our response: Thank you for your considerate comments. Although almost drug regimens are same in doses, some drugs were used at different doses. However, because main interest of our study is to compare the efficacy of pharmacologic interventions used to prevent PONV, we regarded the same drug which was used in different dose and at different time points as same pharmacological strategies and analyzed as one pharmacological strategy. According to reviewer’s comment, we revised the manuscript and provided the information of different doses (low/high doses), time of administration of preventive drugs in Supplementary Table 3.

- Considering preoperative individual and variable risk of each included individuals, any major reason not to consider a Bayesian approach to NMA? You consider such a homogeneous population and may be that is the reason?

Our response: We sincerely appreciate your considerate comments and suggestions for the betterment of our manuscript. When we planned this systematic review and network meta-analysis, we wanted to include homogenous population. Thus, we set the inclusion criteria as narrow as possible. We included only patients who underwent elective ambulatory thyroidectomy under general anesthesia. 

- You included several multi-arm trials. Comparisons within multi‐arm studies are correlated, did you adjust the standard error of each two‐arm comparison to multi‐arm studies. Please consider the method of Rücker and Schwarzer (back‐calculated standard errors in the weighted least‐square estimator).

Our response: As we made a comment above, we applied the methods in developed protocol for this systematic review and network meta-analysis. Thank you.

- Considering combined treatments (e.g. A + B, A + B + C, B + C) are additive sums of their components, why do not present analysis to all mono‐prophylaxis and later to combination prophylaxis? Please argue about your analysis.

Our response: Thank you for reviewer’s comment. As you know, we developed the protocol for this systematic review and network meta-analysis before the beginning of our work. At that stage, we considered and discussed the analysis plan for mono-prophylaxis and combination prophylaxis. At that time, we thought if we planned too many subgroup analysis, it would be too complicated to analyze and read the manuscript. (As you know, this manuscript already has too many analysis, pictures and tables.) And readers may not read our articles. Thus, we decided to analysis all mono‐prophylaxis and combination prophylaxis as each strategy. 

Results.

- In line 261, I believe you are talking about “preoperative seventeen…”. That should be more clear from methods section. You are considering preventing drugs pre or intraop. administered but not postoperative. Please clarify this in detail.

Our response: According to reviewer’s comment, we revised the manuscript. (From line 105 to 106 and from line 109 to 110 on page 6 in the Materials and Methods section).

- Not very sure if “Oral Gin” should be considered a pharmacological active agent. May be need some short clarification in discussion.

Our response: Thank you for your suggestion. At the study selection stage, we were faced with this issue. One investigator include ref 66, but other investigators did not. Thus, discussion was made with a third investigator (H.K.). Firstly, we searched if oral ginger was pharmacologically used. Search showed that oral ginger was pharmacologically used in various situations (Tjendraputra E, Tran VH, Liu-Brennan D, et al. Bioorg Chem 2001; 29:156-163, Altman RD, Marcussen KC. Arthritis Rheum 2001; 44:2531-253, Zhang M, Viennois E, Prasad M, et al. Biomaterials 2016; 101:321-340.) and pharmacokinetic studies for ginger were performed.(Jiang SZ, Wang NS, Mi SQ. Biopharm Drug Dispos 2008; 29:529-537., Zick SM, Djuric Z, Ruffin MT et al, Cancer Epidemiol Biomark Prev 2008; 17:1930-1936.) And we can search a lot of commercially available complementary medicines which contains ginger. And some website introduces ginger capsule as drugs and medications (https://www.webmd.com/drugs/2/drug-21048/ginger-extract-oral/details). Thus, through a discussion, we decided to include that study to our systematic review and network meta-analysis.

- Fig 2a is very clear and describe very well the overall evidence of this population. However, the main node “Con” is not previously described as an intervention. It is referring to Placebo, please clarify. Also, for the text about loops and direct/indirect evidence.

- Considering reference treatment to Con (Placebo) is quite worrying. Did you consider to use any other widely used intervention as reference? let say Dexamethasone?

Our response: Thank you for your considerate and meticulous comments. According to reviewer’s comment, we revised Con to Placebo in the whole manuscript.

Discussion.

- You clearly state that “Its efficacy has been demonstrated when administered for both induction and maintenance anesthesia, but not when given as a bolus dose before the end of surgery for preventing PONV”. However, the overall research process does not clearly address the timing or doses of administered interventions on each trial. Please see above in previous comments.

Our response: Thanks for your comments. That sentence meant the previous studies. And we provide the timing or doses of administered interventions on each trial in Supplementary Table 3. 

- Again, “propofol was efficacious in treating PONV at plasma concentrations that do not produce increased sedation”. Did you consider any AR, Serious AE or any safety related outcomes?

Our response: Thank you for your considerate comments and suggestions for the betterment of the manuscript. As above we mentioned, we provide the table on safety issues (Supplementary Table 2).

- While limitations are clear, why not to suggest exactly what types of comparisons are more needed at large-scale RCT? That should be an interesting finding for further RCTs.

Our response: We sincerely appreciate your comments and suggestion. We revised the manuscript according to reviewer’s comment. However, there are too many kinds of comparisons were not performed. (line from 571 to 572 on page 51 in the Discussion section) Thus, we cannot specify the exact comparison to be conducted. 

- Please explain the reporting bias regarding safety outcomes.

Our response: As we made a comment above, we provide the table on safety issues. (Supplementary Table 2).  

Reviewer #2: This network meta-analysis is to determine the effectiveness of pharmacologic interventions for preventing postoperative nausea and vomiting (PONV) in patients undergoing thyroidectomy. The primary endpoints were the incidences of postoperative nausea (PON), postoperative vomiting (POV), PONV, use of rescue antiemetic and incidence of complete response in the overall postoperative phases. The secondary endpoints were the same parameters assessed in the early, middle, and late postoperative phases. The surface under the cumulative ranking curve (SUCRA) values and rankograms were used to present the hierarchy of pharmacologic interventions.

Twenty-nine studies (n=3,755 patients) that investigated 18 different pharmacologic interventions were included. The incidence of PONV among the overall postoperative phases was lowest with propofol alone (16.1%). The least usage of rescue antiemetics among the overall postoperative phases and the highest complete response was observed with tropisetron and propofol combination (3.9% and 96.6%, respectively). The authors concluded that propofol and tropisetron alone and in combination, and the ramosetron and dexamethasone combination effectively prevented PONV in patients undergoing thyroidectomy, with some heterogeneity observed in this NMA of full-text reports.

This is a detailed network meta-analysis . However, it has several primary outcomes, PON, PON and PONV. Also, there are 18 different pharmacologic interventions. Thus, there are many pages of results, large number of figures, and many tables making this difficult to read and understand the true impact of this network meta-analysis. It will benefit in reduction of the primary end points.

Our response: Thank you for your considerate and meticulous comments. We also agree with reviewer’s comment that too many pages of results, large number of figures, and many tables make this manuscript difficult to read and understand the true impact of this network meta-analysis. Thus, according to editor and reviewer’s comment, we moved the many figures to supplementary file. 

We developed the protocol for this systematic review and network meta-analysis before the beginning of our work, which was registered to the PROSPERO network and also published in a peer reviewed journal. The purpose of these processes was to make the research process transparent and reproducible. Thus, although we totally agree with your opinion, we did not reduce the primary endpoints. Again, thank you very much for your constructive comments. 

As the author pointed out in their objective, the study is to determine the effectiveness of pharmacologic interventions for preventing postoperative nausea and vomiting (PONV) in patients undergoing thyroidectomy. The primary end point is PONV. The authors have chosen to present PON first, then POV, then PONV results. Suggest that the authors presented the PONV data first. For the data of PON and the data on POV, the authors could highlight the key differences in results without going into details. Also some of the tables and some of the figures can be inserted into supplementary materials.

Our response: Thank you for your suggestion. According to reviewer’s suggestion, we revised the order of the outcomes reported throughout the manuscript. We also moved the many figures to supplementary file. 

Ref 66: dexamethasone (Dex)+Oral ginger. The study indicated that it is pharmacological intervention. Does oral ginger consider pharmacological?

Our response: Thank you for reviewer’s suggestion. At the study selection stage, we were faced with this issue. One investigator include ref 66, but other investigator did not. Thus, discussion was made with a third investigator(H.K.). Firstly, we searched if oral ginger was pharmacologically used. Search showed that oral ginger was pharmacologically used in various situations (Tjendraputra E, Tran VH, Liu-Brennan D, et al. Bioorg Chem 2001; 29:156-163, Altman RD, Marcussen KC. Arthritis Rheum 2001; 44:2531-253, Zhang M, Viennois E, Prasad M, et al. Biomaterials 2016; 101:321-340.) and pharmacokinetic studies for ginger were performed.(Jiang SZ, Wang NS, Mi SQ. Biopharm Drug Dispos 2008; 29:529-537., Zick SM, Djuric Z, Ruffin MT et al, Cancer Epidemiol Biomark Prev 2008; 17:1930-1936.) And we can search a lot of commercially available complementary medicines which contains ginger. And some website introduce ginger capsule as drugs and medications (https://www.webmd.com/drugs/2/drug-21048/ginger-extract-oral/details). Thus, through a discussion, we decided to include that study to our systematic review and network meta-analysis.

The postoperative period was divided into the early, middle, late, and overall phases. The early phase was defined as 0–6 h postoperatively; middle phase, 6–24 h postoperatively; and late phase, more than 24 h postoperatively. If a study reported data at multiple time points within the same phase, data from the first time point were selected as the outcome of interest (e.g., if the study reported data at 0 h, 2 h, 4 h, and 6 h postoperatively, we only included the data at 0 h as the early phase). 

This classification may have biased the results towards propofol, since a bolus given at end of surgery may abort PONV immediately, but it may not have sustained effect except if given as infusion.

Our response: Thank you for reviewer’s comment and suggestion. We agree with reviewer’s opinion that the classification used in this systematic review may have biased the results towards propofol. We also agree with reviewer’s opinion that the effect of propofol to prevent PONV not have sustained effect except if given as infusion. 

At the data extraction and statistical analysis stage, we found that all the propofol to prevent PONV are given at the end of surgery, and thought that propofol given at the end of surgery was one of the strategies to prevent PONV after thyroidectomy. Thus, we made a comment for this at the discussion section. However, according to reviewer’s comment, we specify that strategy as propofol, given as a bolus before the end of surgery in the abstract and conclusion section. 

To ensure the inclusion of maximum number of studies, any PON, POV, and PONV data from studies that do not mention a specific time point were defined as the overall phase. What is meant by overall phase, is this 6 to 24h? This may also misclassify groups.

Our response: Thank you for reviewer’s comment. Some studies reported the outcome such as PON, POV or PONV data, in which not specifying the investigating time points. Although these studies did not report the specific time points, PON, POV or PONV data were investigated during the follow-up period of these studies. Thus, we thought those results providing the overall effectiveness of pharmacological interventions for PONV, it was reasonable to include them to this systematic review and network meta-analysis. According to reviewer’s comment, we revised the manuscript to clarify its exact meaning. (line 129 on page 7 in the Materials and Methods section)

We hope the revised manuscript will better meet the requirements of the ‘Plos One’ for publication. Again, we are most grateful for the constructive review by you and members of Editorial Board of the ‘Plos One’.

Sincerely yours,

Hyun Kang, M. D., Ph.D., M.P.H.

Professor

Department of Anaesthesiology and Pain Medicine

Chung-Ang University College of Medicine

224-1 Heukseok-dong, Dongjak-gu

Seoul, 156-755, Korea

Tel: +82-2-6299-2571, 2579, 2586

Mobile:+82-10-8761-4426

Fax: 82-2-6299-2585

E-mail: roman00@naver.com

---

## [Decision Letter · Decision Letter 1]

25 Nov 2020

PONE-D-20-20400R1

Pharmacologic interventions for postoperative nausea and vomiting after thyroidectomy: a systematic review and network meta-analysis

PLOS ONE

Dear Dr. Kang,

Thank you for submitting your manuscript to PLOS ONE. After careful consideration, we feel that it has merit but does not fully meet PLOS ONE’s publication criteria as it currently stands. Therefore, we invite you to submit a revised version of the manuscript that addresses the points raised during the review process.

ACADEMIC EDITOR:

Thanks for submitting your revised version, which has been reviewed by one of the initial reviewers and myself as Editor.

In response to our comments you have provided responses and edits to your manuscript. There are still some points that need to be addressed:

- Thanks for adding a more detailed explanation of the PrI. However, this information requires an appropriate citation.

- Thanks for considering the League tables. Please add footnotes that may explain the readers how to read and interpret them. You may want to use the following paper as an example: https://www.thelancet.com/journals/lancet/article/PIIS0140-6736(17)32802-7/fulltext

We look forward to receiving your revised manuscript.

Kind regards,

Ivan D. Florez

Academic Editor

PLOS ONE

Additional Editor Comments (if provided):

Thanks for submitting your revised version, which has been reviewed by one of the initial reviewers and myself as Editor.

In response to our comments you have provided responses and edits to your manuscript. There are still some points that need to be addressed:

- Thanks for adding a more detailed explanation of the PrI. However, this information requires an appropriate citation.

- Thanks for considering the League tables. Please add footnotes that may explain the readers how to read and interpret them. You may want to use the following paper as an example: https://www.thelancet.com/journals/lancet/article/PIIS0140-6736(17)32802-7/fulltext

Reviewers' comments:

Reviewer's Responses to Questions

**Comments to the Author**

1. If the authors have adequately addressed your comments raised in a previous round of review and you feel that this manuscript is now acceptable for publication, you may indicate that here to bypass the “Comments to the Author” section, enter your conflict of interest statement in the “Confidential to Editor” section, and submit your "Accept" recommendation.

Reviewer #1: All comments have been addressed

2. Is the manuscript technically sound, and do the data support the conclusions?

Reviewer #1: Yes

3. Has the statistical analysis been performed appropriately and rigorously? 

Reviewer #1: Yes

4. Have the authors made all data underlying the findings in their manuscript fully available?

Reviewer #1: Yes

5. Is the manuscript presented in an intelligible fashion and written in standard English?

Reviewer #1: Yes

6. Review Comments to the Author

Reviewer #1: Dear authors,

Thank you for submitting a revised version of your manuscript. This has considerably improved the points under consideration.

7. PLOS authors have the option to publish the peer review history of their article (what does this mean?). If published, this will include your full peer review and any attached files.

Reviewer #1: **Yes: **Jose Andres (JA) Calvache

---

## [Author Response · Author response to Decision Letter 1]

25 Nov 2020

November 25, 2020

Dear Dr Ivan D. Florez, Academic Editor of ‘Plos One’,

Re: Manuscript number PONE-D-20-20400R1, “Pharmacologic interventions for postoperative nausea and vomiting after thyroidectomy: a systematic review and network meta-analysis”

We thank sincerely the academic editor and reviewers of the ‘Plos One’ for taking their precious time to review our paper. According to your comments and suggestions, we sincerely and earnestly tried to response for your comments. We want to express my heartfelt gratitude for your comments once more. 

We have made some corrections in the manuscript after going over your comments. We highlighted the modification made to the original document by using red colored text. The changes are summarized below:

Academic editor’s comment

Thanks for submitting your revised version, which has been reviewed by one of the initial reviewers and myself as Editor.

In response to our comments you have provided responses and edits to your manuscript. There are still some points that need to be addressed:

- Thanks for adding a more detailed explanation of the PrI. However, this information requires an appropriate citation.

Our response: Thank you for editor’s suggestions for the betterment of the manuscript. According to editor’s suggestion, we cited appropriate citation(line 237 on page 12 in the Methods section)

- Thanks for considering the League tables. Please add footnotes that may explain the readers how to read and interpret them. You may want to use the following paper as an example: https://www.thelancet.com/journals/lancet/article/PIIS0140-6736(17)32802-7/fulltext

Our response: Thank you for editor’s suggestions for the betterment of the manuscript. According to editor’s suggestion, we added the foot notes. (Table 3, 4, 5, 6, 7)

We hope the revised manuscript will better meet the requirements of the ‘Plos One’ for publication. Again, we are most grateful for the constructive review by you and members of Editorial Board of the ‘Plos One’.

Sincerely yours,

Hyun Kang, M. D., Ph.D., M.P.H.

Professor

Department of Anaesthesiology and Pain Medicine

Chung-Ang University College of Medicine

224-1 Heukseok-dong, Dongjak-gu

Seoul, 156-755, Korea

Tel: +82-2-6299-2571, 2579, 2586

Mobile:+82-10-8761-4426

Fax: 82-2-6299-2585

E-mail: roman00@naver.com

---

## [Editor Report · Decision Letter 2]

30 Nov 2020

Pharmacologic interventions for postoperative nausea and vomiting after thyroidectomy: a systematic review and network meta-analysis

PONE-D-20-20400R2

Dear Dr. Kang,

We’re pleased to inform you that your manuscript has been judged scientifically suitable for publication and will be formally accepted for publication once it meets all outstanding technical requirements.

Kind regards,

Ivan D. Florez

Academic Editor

PLOS ONE
---

## [Editor Report · Acceptance letter]

4 Dec 2020

PONE-D-20-20400R2 

Pharmacologic interventions for postoperative nausea and vomiting after thyroidectomy: a systematic review and network meta-analysis 

Dear Dr. Kang:

I'm pleased to inform you that your manuscript has been deemed suitable for publication in PLOS ONE. Congratulations! Your manuscript is now with our production department. 

Kind regards, 

on behalf of

Dr. Ivan D. Florez 

Academic Editor

PLOS ONE